# LINKING AVERAGE- AND WORST-CASE PERTURBATION ROBUSTNESS VIA CLASS SELECTIVITY AND DIMENSIONALITY

## ABSTRACT

Representational sparsity is known to affect robustness to input perturbations in deep neural networks (DNNs), but less is known about how the semantic content of representations affects robustness. Class selectivity—the variability of a unit's responses across data classes or dimensions—is one way of quantifying the sparsity of semantic representations. Given recent evidence that class selectivity may not be necessary for, and in some cases can impair generalization, we sought to investigate whether it also confers robustness (or vulnerability) to perturbations of input data. We found that class selectivity leads to increased vulnerability to average-case (naturalistic) perturbations in ResNet18, ResNet50, and ResNet20, as measured using Tiny ImageNetC (ResNet18 and ResNet50) and CIFAR10C (ResNet20). Networks regularized to have lower levels of class selectivity are more robust to average-case perturbations, while networks with higher class selectivity are more vulnerable. In contrast, we found that class selectivity increases robustness to multiple types of worst-case (i.e. white box adversarial) perturbations, suggesting that while decreasing class selectivity is helpful for average-case perturbations, it is harmful for worst-case perturbations. To explain this difference, we studied the dimensionality of the networks' representations: we found that the dimensionality of early-layer representations is inversely proportional to a network's class selectivity, and that adversarial samples cause a larger increase in early-layer dimensionality than corrupted samples. We also found that the input-unit gradient was more variable across samples and units in high-selectivity networks compared to low-selectivity networks. These results lead to the conclusion that units participate more consistently in low-selectivity regimes compared to high-selectivity regimes, effectively creating a larger attack surface and hence vulnerability to worst-case perturbations.

## 1 INTRODUCTION

Methods for understanding deep neural networks (DNNs) often attempt to find individual neurons or small sets of neurons that are representative of a network's decision (Erhan et al., 2009; Zeiler and Fergus, 2014; Karpathy et al., 2016; Amjad et al., 2018; Lillian et al., 2018; Dhamdhere et al., 2019; Olah et al., 2020). Selectivity in individual units (i.e. variability in a neuron's activations across semantically-relevant data features) has been of particular interest to researchers trying to better understand deep neural networks (DNNs) (Zhou et al., 2015; Olah et al., 2017; Morcos et al., 2018; Zhou et al., 2018; Meyes et al., 2019; Na et al., 2019; Zhou et al., 2019; Rafegas et al., 2019; Bau et al., 2020; Leavitt and Morcos, 2020). However, recent work has shown that selective neurons can be irrelevant, or even detrimental to network performance, emphasizing the importance of examining distributed representations for understanding DNNs (Morcos et al., 2018; Donnelly and Roegiest, 2019; Dalvi et al., 2019b; Leavitt and Morcos, 2020).

In parallel, work on robustness seeks to build models that are robust to perturbed inputs (Szegedy et al., 2013; Carlini and Wagner, 2017a;b; Vasiljevic et al., 2016; Kurakin et al., 2017; Gilmer et al., 2018; Zheng et al., 2016). Hendrycks and Dietterich (2019) distinguish between two types of robustness: corruption robustness, which measures a classifier's performance on low-quality or naturalistically-perturbed inputs—and thus is an "average-case" measure—and adversarial robustness,

which measures a classifier's performance on small, additive perturbations that are tailored to the classifier—and thus is a "worst-case" measure.[1]

Research on robustness has been predominantly focused on worst-case perturbations, which is affected by weight and activation sparsity (Madry et al., 2018; Balda et al., 2020; Ye et al., 2018; Guo et al., 2018; Dhillon et al., 2018) and representational dimensionality (Langeberg et al., 2019; Sanyal et al., 2020; Nayebi and Ganguli, 2017). But less is known about the mechanisms underlying average-case perturbation robustness and its common factors with worst-case robustness. Some techniques for improving worst-case robustness also improve average-case robustness (Hendrycks and Dietterich, 2019; Ford et al., 2019; Yin et al., 2019), thus it is possible that sparsity and representational dimensionality also contribute to average-case robustness. Selectivity in individual units can be also be thought of a measure of the sparsity with which semantic information is represented.[2] And because class selectivity regularization provides a method for controlling selectivity, and class selectivity regularization has been shown to improve test accuracy on unperturbed data (Leavitt and Morcos, 2020), we sought to investigate whether it could be utilized to improve perturbation robustness and elucidate the factors underlying it.

In this work we pursue a series of experiments investigating the causal role of selectivity in robustness to worst-case and average-case perturbations in DNNs. To do so, we used a recently-developed class selectivity regularizer (Leavitt and Morcos, 2020) to directly modify the amount of class selectivity learned by DNNs, and examined how this affected the DNNs' robustness to worst-case and average-case perturbations. Our findings are as follows:

- Networks regularized to have lower levels of class selectivity are more robust to average-case perturbations, while networks with higher class selectivity are generally less robust to average-case perturbations, as measured in ResNets using the Tiny ImageNetC and CIFAR10C datasets. The corruption robustness imparted by regularizing against class selectivity was consistent across nearly all tested corruptions.
- In contrast to its impact on average-case perturbations, decreasing class selectivity *reduces* robustness to worst-case perturbations in both tested models, as assessed using gradient-based white-box attacks.
- The variability of the input-unit gradient across samples and units is proportional to a network's overall class selectivity, indicating that high variability in perturbability within and across units may facilitate worst-case perturbation robustness.
- The dimensionality of activation changes caused by corruption markedly increases in early layers for both perturbation types, but is larger for worst-case perturbations and low-selectivity networks. This implies that representational dimensionality may present a trade-off between worst-case and average-case perturbation robustness.

Our results demonstrate that changing class selectivity, and hence the sparsity of semantic representations, can confer robustness to average-case or worst-case perturbations, but not both simultaneously. They also highlight the roles of input-unit gradient variability and representational dimensionality in mediating this trade-off.

## 2 RELATED WORK

### 2.1 PERTURBATION ROBUSTNESS

The most commonly studied form of robustness in DNNs is robustness to adversarial attacks, in which an input is perturbed in a manner that maximizes the change in the network's output while

---

[1]We use the terms "worst-case perturbation" and "average-case perturbation" instead of "adversarial attack" and "corruption", respectively, because this usage is more general and dispenses with the implied categorical distinction of using seemingly-unrelated terms. Also note that while Hendrycks and Dietterich (2019) assign specific and distinct meanings to "perturbation" and "corruption", we use the term "perturbation" more generally to refer to any change to an input.

[2]Class information is semantic. And because class selectivity measures the degree to which class information is represented in *individual* neurons, it can be considered a form of sparsity. For example, if a network has high test accuracy on a classification task, it is necessarily representing class (semantic) information. But if the mean class selectivity across units is low, then the individual units do not contain much class information, thus the class information must be distributed across units; the semantic representation in this case is not sparse, it is distributed.

attempting to minimize or maintain below some threshold the magnitude of the change to the input (Serban et al., 2019; Warde-Farley and Goodfellow, 2017) . Because white-box adversarial attacks are optimized to best confuse a given network, robustness to adversarial attacks are a "worst-case" measure of robustness. Two factors that have been proposed to account for DNN robustness to worst-case perturbations are particularly relevant to the present study: sparsity and dimensionality.

Multiple studies have linked activation and weight sparsity with robustness to worst-case perturbations. Adversarial training improves worst-case robustness Goodfellow et al. (2015); Huang et al. (2016) and results in sparser weight matrices (Madry et al., 2018; Balda et al., 2020). Methods for increasing the sparsity of weight matrices (Ye et al., 2018; Guo et al., 2018) and activations (Dhillon et al., 2018) likewise improve worst-case robustness, indicating that the weight sparsity caused by worst-case perturbation training is not simply a side-effect.

Researchers have also attempted to understand the nature of worst-case robustness from a perspective complementary to that of sparsity: dimensionality. Like sparsity, worst-case perturbation training reduces the rank of weight matrices and representations, and regularizing weight matrices and representations to be low-rank can improve worst-case perturbation robustness (Langeberg et al., 2019; Sanyal et al., 2020; Nayebi and Ganguli, 2017). Taken together, these studies support the notion that networks with low-dimensional representations are more robust to worst-case perturbations.

Comparatively less research has been conducted to understand the factors underlying average-case robustness. Certain techniques for improving worst-case perturbation robustness also help against average-case perturbations (Hendrycks and Dietterich, 2019; Geirhos et al., 2018; Ford et al., 2019). Examining the frequency domain has elucidated one mechanism: worst-case perturbations for "baseline" models tend to be in the high frequency domain, and improvements in average-case robustness resulting from worst-case robustness training are at least partially ascribable to models becoming less reliant on high-frequency information (Yin et al., 2019; Tsuzuku and Sato, 2019; Geirhos et al., 2018). But it remains unknown whether other factors such as sparsity and dimensionality link these two forms of robustness.

## 2.2 CLASS SELECTIVITY

One technique that has been of particular interest to researchers trying to better understand deep (and biological) neural networks is examining the selectivity of individual units (Zhou et al., 2015; Olah et al., 2017; Morcos et al., 2018; Zhou et al., 2018; Meyes et al., 2019; Na et al., 2019; Zhou et al., 2019; Rafegas et al., 2019; Bau et al., 2020; Leavitt and Morcos, 2020; Sherrington, 1906; Kandel et al., 2000). Evidence regarding the importance of selectivity has mostly relied on single unit ablation, and has been equivocal (Radford et al., 2017; Morcos et al., 2018; Amjad et al., 2018; Zhou et al., 2018; Donnelly and Roegiest, 2019; Dalvi et al., 2019a). However Leavitt and Morcos (2020) examined the role of single unit selectivity in network performance by regularizing for or against class selectivity in the loss function, which sidesteps the limitations of single unit ablation and correlative approaches and allowed them to investigate the causal effect of class selectivity. They found that reducing class selectivity has little negative impact on—and can even improve—test accuracy in CNNs trained on image recognition tasks, but that increasing class selectivity has significant negative effects on test accuracy. However, their study focused on examining the effects of class selectivity on test accuracy in unperturbed (clean) inputs. Thus it remains unknown how class selectivity affects robustness to perturbed inputs, and whether class selectivity can serve as or elucidate a link between worst-case and average-case robustness.

## 3 APPROACH

A detailed description of our approach is provided in Appendix A.1.

**Models and training protocols**   Our experiments were performed on ResNet18 and ResNet50 (He et al., 2016) trained on Tiny ImageNet (Fei-Fei et al., 2015), and ResNet20 (He et al., 2016) trained on CIFAR10 (Krizhevsky, 2009). We focus primarily on the results for ResNet18 trained on Tiny ImageNet in the main text for space, though results were qualitatively similar for ResNet50, and ResNet20 trained on CIFAR10. Experimental results were obtained with model parameters from the epoch that achieved the highest validation set accuracy over the training epochs, and 20 replicate

models (ResNet18 and ResNet20) or 5 replicate models (Resnet50) with different random seeds were run for each hyperparameter set.

**Class selectivity index**   Following (Leavitt and Morcos, 2020). A unit's class selectivity index is calculated as follows: At every ReLU, the activation in response to a single sample was averaged across all elements of the filter map (which we refer to as a "unit"). The class-conditional mean activation was then calculated across all samples in the clean test set, and the class selectivity index ($SI$) was calculated as follows:

$$SI = \frac{\mu_{max} - \mu_{-max}}{\mu_{max} + \mu_{-max}} \tag{1}$$

where $\mu_{max}$ is the largest class-conditional mean activation and $\mu_{-max}$ is the mean response to the remaining (i.e. non-$\mu_{max}$) classes. The selectivity index ranges from 0 to 1. A unit with identical average activity for all classes would have a selectivity of 0, and a unit that only responds to a single class would have a selectivity of 1.

As Morcos et al. (2018) note, the selectivity index is not a perfect measure of information content in single units. For example, a unit with a litte bit of information about many classes would have a low selectivity index. However, it identifies units that are class-selective similarly to prior studies (Zhou et al., 2018). Most importantly, it is differentiable with respect to the model parameters.

**Class selectivity regularization**   We used (Leavitt and Morcos, 2020)'s class selectivity regularizer to control the levels of class selectivity learned by units in a network during training. Class selectivity regularization is achieved by minimizing the following loss function during training:

$$loss = -\sum_{c}^{C} y_c \cdot \log(\hat{y}_c) - \alpha \mu_{SI} \tag{2}$$

The left-hand term in the loss function is the standard classification cross-entropy, where $c$ is the class index, $C$ is the number of classes, $y_c$ is the true class label, and $\hat{y}_c$ is the predicted class probability. The right-hand component of the loss function, $-\alpha \mu_{SI}$, is the class selectivity regularizer. The regularizer consists of two terms: the selectivity term,

$$\mu_{SI} = \frac{1}{L} \sum_{l}^{L} \frac{1}{U} \sum_{u}^{U} SI_{u,l} \tag{3}$$

where $l$ is a convolutional layer, $L$ is number of layers, $u$ is a unit, $U$ is the number of units in a given layer, and $SI_u$ is the class selectivity index of unit $u$. The selectivity term of the regularizer is obtained by computing the selectivity index for each unit in a layer, then computing the mean selectivity index across units within each layer, then computing the mean selectivity index across layers. Computing the mean within layers before computing the mean across layers (as compared to computing the mean across all units in the network) mitigates the biases induced by the larger numbers of units in deeper layers. The other term in the regularizer is $\alpha$, the regularization scale, which determines whether class selectivity is promoted or discouraged. Negative values of $\alpha$ discourage class selectivity in individual units and positive values encourage it. The magnitude of $\alpha$ controls the contribution of the selectivity term to the overall loss. During training, the class selectivity index was computed for each minibatch. The final (logit) layer was not subject to selectivity regularization or included in our analyses because by definition, the logit layer must be class selective in a classification task.

**Measuring average-case robustness**   To evaluate robustness to average-case perturbations, we tested our networks on CIFAR10C and Tiny ImageNetC, two benchmark datasets consisting of the CIFAR10 or Tiny ImageNet data, respectively, to which a set of naturalistic corruptions have been applied (Hendrycks and Dietterich, 2019, examples in Figure A1). We average across all corruption types and severities (see Appendix A.1.2 for details) when reporting corrupted test accuracy.

**Measuring worst-case robustness**   We tested our models' worst-case (i.e. adversarial) robustness using two methods. The fast gradient sign method (FGSM) (Goodfellow et al., 2015) is a simple attack that computes the gradient of the loss with respect to the input image, then scales the image's pixels (within some bound) in the direction that increases the loss. The second method, projected gradient descent (PGD) (Kurakin et al., 2016; Madry et al., 2018), is an iterated version of FGSM. We used a step size of 0.0001 and an $l_\infty$ norm perturbation budget ($\epsilon$) of 16/255.

**Computing the stability of units and layers**   To quantify variation in networks' perturbability, we first computed the $l_2$ norm of the input-unit gradient for each unit $u$ in a network. We then computed the mean ($\mu_u$) and standard deviation ($\sigma_u$) of the norm across samples for each unit. $\sigma_u/\mu_u$ yields

**a)**

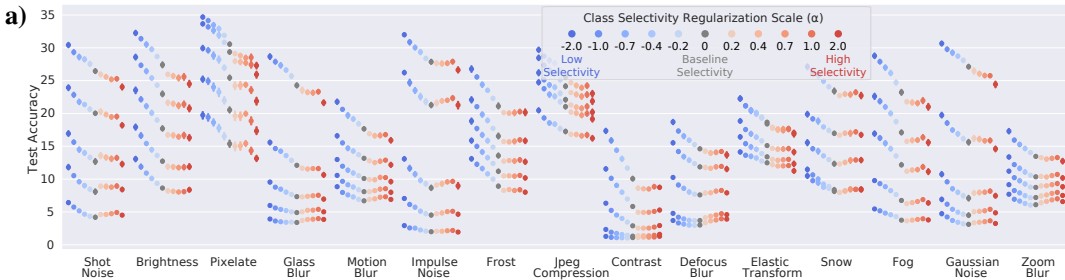

**b)**

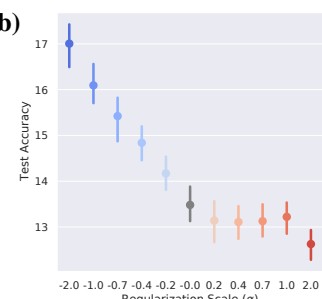

**Figure 1: Reducing class selectivity improves average-case robustness.** Test accuracy (y-axis) as a function of corruption type (x-axis), class selectivity regularization scale ($\alpha$; color), and corruption severity (ordering along y-axis). Test accuracy is reduced proportionally to corruption severity, leading to an ordering along the y-axis; corruption severity 1 (least severe) is at the top, corruption severity 5 (most severe) at the bottom. (**b**) Mean test accuracy across all corruptions and severities (y-axis) as a function of $\alpha$ (x-axis). Results shown are for ResNet18 trained on Tiny ImageNet and tested on Tiny ImageNetC. Error bars = 95% confidence intervals of the mean. See Figure A6 for CIFAR10C results.

the coefficient of variation (Everitt, 2002) for a unit ($CV_u$), a measure of variation in perturbability for individual units. We also quantified the variation *across* units in a layer by computing the standard deviation of $\mu_u$ across units in a layer $l$, $\sigma(\mu_u) = \sigma_l$, and dividing this by the corresponding mean across units $\mu(\mu_u) = \mu_l$, to yield the CV across units $\sigma_l/\mu_l = CV_l$.

## 4 RESULTS

### 4.1 AVERAGE-CASE ROBUSTNESS IS INVERSELY PROPORTIONAL TO CLASS SELECTIVITY

Certain kinds of sparsity—including reliance on single directions (Morcos et al., 2018), and the semantic sparsity measured by class selectivity (Leavitt and Morcos, 2020)—have been shown to impair network performance. We sought to extend this question to robustness: how does the sparsity of semantic representations affect robustness to average-case perturbations of the input data? We used a recently-introduced method (Leavitt and Morcos (2020); Approach 3) to modulate the amount of class selectivity learned by DNNs (Figure A2 demonstrates effects of selectivity regularization). We then examined how this affected performance on Tiny ImageNetC and CIFAR10C, two benchmark datasets for average-case corruptions (Approach 3; example images in Figure A1).

Changing the level of class selectivity across neurons in a network could one of the following effects on corruption robustness: If concentrating semantic representations into fewer neurons (i.e. promoting semantic sparsity) provides fewer potent dimensions on which perturbed inputs can act, then increasing class selectivity should confer networks with robustness to average-case perturbations, while reducing class selectivity should render networks more vulnerable. Alternatively, if distributing semantic representations across more units (i.e. reducing sparsity) dilutes the changes induced by perturbed inputs, then reducing class selectivity should increase a network's robustness to average-case perturbations, while increasing class selectivity should reduce robustness.

We found that decreasing class selectivity leads to increased robustness to average-case perturbations for both ResNet18 tested on Tiny ImageNetC (Figure 1) and ResNet20 tested on CIFAR10C (Figure A6). In ResNet18, we found that mean test accuracy on corrupted inputs increases as class selectivity decreases (Figure 1), with test accuracy reaching a maximum at regularization scale $\alpha = -2.0$ (mean test accuracy across corruptions and severities at $\alpha_{-2.0} = 17$), representing a 3.5 percentage point (pp) increase relative to no selectivity regularization (i.e. $\alpha_0$; test accuracy at $\alpha_0 = 13.5$). In contrast, regularizing to increase class selectivity has either no effect or a negative impact on corruption robustness. Corrupted test accuracy remains relatively stable until $\alpha = 1.0$, after which point it declines. The results are qualitatively similar for ResNet50 tested on Tiny ImageNetC (Figure A9), and for ResNet20 tested on CIFAR10C (Figure A6), except the vulnerability to corruption caused by increasing selectivity is even more dramatic in ResNet20. We also found similar results when controlling for the difference in clean accuracy for models with different $\alpha$ (Appendix A.3).

We observed that regularizing to decrease class selectivity causes robustness to average-case perturbations. But it's possible that the causality is unidirectional, leading to the question of whether the

**Figure 2: Class selectivity imparts worst-case perturbation robustness.** (**a**) Test accuracy (y-axis) as a function of perturbation intensity ($\epsilon$; x-axis) and class selectivity regularization scale ($\alpha$; color) for the FGSM attack. (**b**) Test accuracy (y-axis) as a function of attack optimization steps (x-axis) and $\alpha$ for the PGD attack. (**c**) Network stability, as measured with $l_2$ norm of the input-output Jacobian (y-axis), as a function of $\alpha$ (x-axis). All results are for ResNet18 trained on Tiny ImageNet. Shaded region and bars = 95% confidence interval of the mean. See Figure A12 for ResNet20 results.

converse is also true: does increasing robustness to average-case perturbations cause class selectivity to decrease? We investigated this question by training with AugMix, a technique known to improve worst-case robustness (Hendrycks et al., 2020a). We found that AugMix does indeed decrease the mean level of class selectivity across neurons in a network (Appendix A.4; Figure A11). AugMix decreases overall levels of selectivity similarly to training with a class selectivity regularization scale of approximately $\alpha = -0.1$ or $\alpha = -0.2$ in both ResNet18 trained on Tiny ImageNet (Figures A11a and A11b) and ResNet20 trained on CIFAR10 (Figures A11c and A11d). These results indicate that the causal relationship between average-case perturbation robustness and class selectivity is bidirectional: not only does decreasing class selectivity improve average-case perturbation robustness, but improving average-case perturbation-robustness also causes class selectivity to decrease.

We also found that the effect of class selectivity on perturbed robustness is consistent across corruption types. Regularizing against selectivity improves perturbation robustness in all 15 Tiny ImageNetC corruption types for ResNet18 (Figure A4) and 14 of 15 Tiny ImageNetC corruption types in ResNet50 (Figure A10), and 14 of 19 corruption types in CIFAR10C for ResNet20 (Figure A7). Together these results demonstrate that reduced class selectivity confers robustness to average-case perturbations, implying that distributing semantic representations across neurons—i.e. low sparsity—may dilute the changes induced by average-case perturbations.

### 4.2 CLASS SELECTIVITY IMPARTS WORST-CASE PERTURBATION ROBUSTNESS

We showed that the sparsity of a network's semantic representations, as measured with class selectivity, is causally related to a network's robustness to average-case perturbations. But how does the sparsity of semantic representations affect *worst-case* robustness? We addressed this question by testing our class selectivity-regularized networks on inputs that had been perturbed using using one of two gradient-based methods (see Approach 3).

If distributing semantic representations across units provides more dimensions upon which a worst-case perturbation is potent, then worst-case perturbation robustness should be proportional to class selectivity. However, if increasing the sparsity of semantic representations creates more responsive individual neurons, then worst-case robustness should be inversely proportional to class selectivity.

Unlike average-case perturbations, decreasing class selectivity *decreases* robustness to worst-case perturbations for ResNet18 (Figure 2) and ResNet50 (Figure A13) trained on Tiny ImageNet, and ResNet20 trained on CIFAR10 (Figures A12). For small perturbations (i.e. close to x=0), the effects of class selectivity regularization on test accuracy (class selectivity is inversely correlated with unperturbed test accuracy) appear to overwhelm the effects of perturbations. But as the magnitude of perturbation increases, a stark ordering emerges: test accuracy monotonically decreases as a function of class selectivity in ResNet18 and ResNet50 for both FGSM and PGD attacks (ResNet18: Figures 2a and 2b; ResNet50: Figures A13a and A13b). The ordering is also present for ResNet20, though less consistent for the two networks with the highest class selectivity ($\alpha = 0.7$ and $\alpha = 1.0$). However, increasing class selectivity is much more damaging to test accuracy in ResNet20 trained on CIFAR10 compared to ResNet18 trained on Tiny ImageNet (Leavitt and Morcos, 2020, Figure A2), so the the substantial performance deficits of extreme selectivity in ResNet20 likely mask the perturbation-robustness. This result demonstrates that networks with sparse semantic representations are less vulnerable to worst-case perturbation than networks with distributed semantic representations. We also verified that the worst-case robustness of high-selectivity networks is not fully explained by gradient-masking (Athalye et al., 2018, Appendix A.5).

Interestingly, class selectivity regularization does not appear to affect robustness to "natural" adversarial examples (Appendix A.6), which are "unmodified, real-world examples...selected to cause a

model to make a mistake" (Hendrycks et al., 2020b). Performance on ImageNet-A, a benchmark of natural adversarial examples (Hendrycks et al., 2020b), was similar across all tested values of $\alpha$ for both ResNet18 (Figure A15a) and ResNet50 (Figure A15b), indicating that class selectivity regularization may share some limitations with other methods for improving both worst-case and average-case robustness, many of which also fail to yield significant robustness improvements against ImageNet-A (Hendrycks et al., 2020b).

We found that regularizing to increase class selectivity causes robustness to worst-case perturbations. But is the converse true? Does increasing robustness to worst-case perturbations also cause class selectivity to increase? We investigated this by training networks with a commonly-used technique to improve worst-case perturbation robustness, PGD training. We found that PGD training does indeed increase the mean level of class selectivity across neurons in a network, and this effect is proportional to the strength of PGD training: networks trained with more strongly-perturbed samples have higher class selectivity (Appendix A.7). This effect was present in both ResNet18 trained on Tiny ImageNet (Figure A16c) and ResNet20 trained on CIFAR10 (Figure A16f), indicating that the causal relationship between worst-case perturbation robustness and class selectivity is bidirectional.

Networks whose outputs are more stable to small input perturbations are known to have improved generalization performance and worst-case perturbation robustness (Drucker and Le Cun, 1992; Novak et al., 2018; Sokolic et al., 2017; Rifai et al., 2011; Hoffman et al., 2019). To examine whether increasing class selectivity improves worst-case perturbation robustness by increasing network stability, we analyzed each network's input-output Jacobian, which is proportional to its stability—a large-magnitude Jacobian means that a small change to the network's input will cause a large change to its output. If class selectivity induces worst-case robustness by increasing network stability, then networks with higher class selectivity should have smaller Jacobians. But if increased class selectivity induces adversarial robustness through alternative mechanisms, then class selectivity should have no effect on the Jacobian. We found that the $l_2$ norm of the input-output Jacobian is inversely proportional to class selectivity for ResNet18 (Figure 2c), ResNet50 (Figure A13c), and ResNet20 (Figure A12c), indicating that distributed semantic representations are more vulnerable to worst-case perturbation because they are less stable than sparse semantic representations.

### 4.3 Variability of the input-unit gradient across samples and units

We observed that the input-output Jacobian is proportional to worst-case vulnerability and inversely proportional to class selectivity, but focusing on input-output stability potentially overlooks phenomena present in hidden layers and units. If class selectivity imparts worst-case robustness by making individual units less reliably perturbable—because each unit is highly tuned to a particular subset of images—then we should expect to see more variation across input-unit gradients for units in high-selectivity networks compared to units in low-selectivity networks. Alternatively, worst-case robustness in high-selectivity networks could be achieved by reducing both the magnitude and variation of units' perturbability, in which case we would expect to observe lower variation across input-unit gradients for units in high-selectivity networks compared to low-selectivity networks.

We quantified variation in unit perturbability using the coefficient of variation of the input-unit gradient across samples for each unit ($CV_u$; Approach 3). The CV is a measure of variability that normalizes the standard deviation of a quantity by the mean. A large CV indicates high variability, a small CV indicates low variability. To quantify variation in perturbability *across* units, we computed the CV across units in each layer, ($CV_l$; Approach 3).

We found that units in high-selectivity networks exhibited greater variation in their perturbability than units in low-selectivity networks, both within individual units and across units in each layer. This effect was present in both ResNet18 trained on Tiny ImageNet (Figure 3) and ResNet20 trained on CIFAR10 (Figure A18), although the effect was less consistent for across-unit variability in later layers in ResNet18 (Figure 3b). Interestingly, class selectivity affects both the numerator ($\sigma$) and denominator ($\mu$) of the CV calculation for both the CV across samples and CV across units (Appendix A.8). These results indicate that that high class selectivity imparts worst-case robustness by increasing the variation in perturbability within and across units, while the worst-case vulnerability associated with low class selectivity results from more consistently perturbable units. It is worth noting that the inverse can be stated with regards to average-case robustness: low variation in perturbability both within and across units in low-selectivity networks is associated with robustness to average-case perturbations, despite the these units (and networks) being more perturbable on average.

## 4.4 DIMENSIONALITY IN EARLY LAYERS PREDICTS PERTURBATION VULNERABILITY

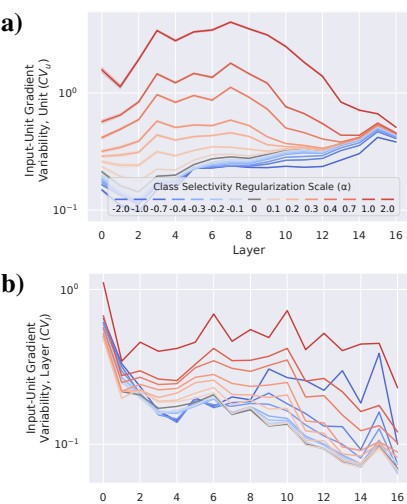

**Figure 3: Class selectivity causes higher variation in perturbability within and across units** (a) Coefficient of variation of input-unit gradient for each unit ($CV_u$; see Approach 3; y-axis) as a function of layer (x-axis). (b) CV of input-unit gradient across units in a layer ($CV_l$; y-axis) as a function of layer (x-axis). Results shown are for ResNet18 trained on Tiny ImageNet. Shaded regions = 95% confidence intervals of the mean. See Appendix A.8 for results for ResNet20 trained on CIFAR10.

Prior research has elucidated the mechanisms of worst-case perturbations using the framework of dimensionality (Langeberg et al., 2019; Nayebi and Ganguli, 2017; Sanyal et al., 2020). Investigating the dimensionality of the changes induced by average-case and worst-case perturbations could reveal a common factor linking them to each other and to class selectivity.

One possible explanation for the discrepancy between class selectivity's impact on worst-case and average-case corruptions is that different corruption types impact representations with varying dimensionalities. For example, if only a few neurons are needed to change a network's decision, the dimensionality of the change in the representation due to perturbation might be very low, as only a few units need to be modified. We thus measured dimensionality using a straightforward, linear method: we applied Principal Component Analysis (PCA) to the activation matrices of each layer in our networks and computed the number of components necessary to explain 95% of the variance (and replicated our results with different variance thresholds; Appendix A.1.4; Appendix A.9). We first examined the dimensionality of the representations of the clean test data. If the sparsity of semantic representations is reflected in dimensionality, then networks with more class selectivity should have lower-dimensional representations than networks with less class selectivity. Alternatively, if high-selectivity representations are of similar dimensionality to low-selectivity representations—though perhaps occupying different sub-spaces—then dimensionality would be unaffected by class selectivity.

We found that the sparsity of a DNN's semantic representations corresponds directly to the dimensionality of those representations. Dimensionality is inversely proportional to class selectivity in early ResNet18 layers ($\leq$layer 9; Figure 4a), and across all of ResNet20 (Figure A21d). Networks with higher class selectivity tend to have lower dimensionality, and networks with lower class selectivity tend to have higher dimensionality. These results show that the sparsity of a network's semantic representations is indeed reflected in those representations' dimensionality.

We next examined the dimensionality of perturbation-induced changes in representations by subtracting the perturbed activation matrix from the clean activation matrix and computing the dimensionality of this "difference matrix" (see Appendix A.1.4). Intuitively, this metric quantifies the dimensionality of the change in the representation caused by perturbing the input. If it is small, the perturbation impacts fewer units, while if it is large, more units are impacted. Interestingly, we found that the dimensionality of the changes in activations induced by both average-case (Figure 4b) and worst-case perturbations (Figure 4c) was notably higher for networks with reduced class-selectivity, suggesting that decreasing class selectivity causes changes in input to become more distributed.

We found that the activation changes caused by average-case perturbations are higher-dimensional than the representations of the clean data in both ResNet18 (compare Figures 4b and 4a) and ResNet20 (Figures A21e and A21d), and that this effect is inversely proportional to class selectivity (Figures 4b and A21e); the increase in dimensionality from average-case perturbations was more pronounced in low-selectivity networks than in high-selectivity networks. These results indicate that class selectivity not only predicts the dimensionality of a representation, but also the change in dimensionality induced by an average-case perturbation.

Notably, however, the increase in early-layer dimensionality was much larger for worst-case perturbations than average-case perturbations (Figure 4c; Figure A21f) . These results indicate that, while the changes in dimensionality induced by both naturalistic and adversarial perturbations are proportional

**Figure 4: Dimensionality in early layers predicts worst-case vulnerability.** (**a**) Fraction of dimensionality (y-axis; see Appendix A.1.4) as a function of layer (x-axis). (**b**) Dimensionality of difference between clean and average-case perturbation activations (y-axis) as a function of layer (x-axis). (**c**) Dimensionality of difference between clean and worst-case perturbation activations (y-axis) as a function of layer (x-axis). Results shown are for ResNet18 trained on Tiny ImageNet. See Appendix A21 for ResNet20.

to the dimensionality of the network's representations, these changes do not consistently project onto coding-relevant dimensions of the representations. Indeed, the larger change in early-layer dimensionality caused by worst-case perturbations likely reflects targeted projection onto coding-relevant dimensions and provides intuition as to why low-selectivity networks are more susceptible to worst-case perturbations.

Hidden layer representations in DNNs are known to lie on non-linear manifolds that are of lower dimensionality than the space in which they're embedded (Goodfellow et al., 2016; Ansuini et al., 2019). Consequently, linear methods such as PCA can provide misleading estimates of hidden layer dimensionality. Thus we also quantified the intrinsic dimensionality (ID) of each layer's representations (see Appendix A.1.4). Interestingly, the results were qualitatively similar to what we observed when examining linear dimensionality (Figure A22) in both ResNet18 trained on Tiny ImageNet (Figure A22a-A22c) and ResNet20 trained on CIFAR10 (Figure A22d-A22f). Thus both linear and non-linear measures of dimensionality imply that representational dimensionality may present a trade-off between worst-case and average-case perturbation robustness.

## 5 DISCUSSION

Our results demonstrate that changes in the sparsity of semantic representations, as measured with class selectivity, induce a trade-off between robustness to average-case vs. worst-case perturbations: highly-distributed semantic representations confer robustness to average-case perturbations, but their increased dimensionality and consistent perturbability result in vulnerability to worst-case perturbations. In contrast, sparse semantic representations yield low-dimensional representations and inconsistently-perturbable units, imparting worst-case robustness. Furthermore, the dimensionality of the difference in early-layer activations between clean and perturbed samples is larger for worst-case perturbations than for average-case perturbations. More generally, our results link average-case and worst-case perturbation robustness through class selectivity and representational dimensionality.

We hesitate to generalize too broadly about our findings, as they are limited to CNNs trained on image classification tasks. It is possible that the results we report here are specific to our models and/or datasets, and also may not extend to other tasks. Scaling class selectivity regularization to datasets with large numbers of classes also remains an open problem (Leavitt and Morcos, 2020).

Our findings could be utilized for practical ends and to clarify findings in prior work. Relevant to both of these issues is the task of adversarial example detection. There is conflicting evidence that intrinsic dimensionality can be used to characterize or detect adversarial (worst-case) samples (Ma et al., 2018; Lu et al., 2018). The finding that worst-case perturbations cause a marked increase in both intrinsic and linear dimensionality indicates that there may be merit in continuing to study these quantities for use in worst-case perturbation detection. And the observation that the causal relationship between class-selectivity and worst- and average-case robustness is bidirectional helps clarify the known benefits of sparsity (Madry et al., 2018; Balda et al., 2020; Ye et al., 2018; Guo et al., 2018; Dhillon et al., 2018) and dimensionality (Langeberg et al., 2019; Sanyal et al., 2020; Nayebi and Ganguli, 2017) on worst-case robustness. It furthermore raises the question of whether enforcing low-dimensional representations also causes class selectivity to increase.

Our work may also hold practical relevance to developing robust models: class selectivity could be used as both a metric for measuring model robustness and a method for achieving robustness (via regularization). We hope future work will more comprehensively assess the utility of class selectivity as part of the deep learning toolkit for these purposes.

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

# A APPENDIX

## A.1 DETAILED APPROACH

Unless otherwise noted: all experimental results were derived from the corrupted or adversarial test set with the parameters from the epoch that achieved the highest clean validation set accuracy over the training epochs; 20 replicates with different random seeds were run for each hyperparameter set; error bars and shaded regions denote bootstrapped 95% confidence intervals; selectivity regularization was not applied to the final (output) layer, nor was the final layer included in any of our analyses.

### A.1.1 MODELS

All models were trained using stochastic gradient descent (SGD) with momentum = 0.9 and weight decay = 0.0001. The maxpool layer after the first batchnorm layer in ResNet18 (see He et al. (2016)) was removed because of the smaller size of Tiny ImageNet images compared to standard ImageNet images (64x64 vs. 256x256, respectively). ResNet18 and ResNet50 were trained for 90 epochs with a minibatch size of 4096 (ResNet18) or 1400 (ResNet50) samples with a learning rate of 0.1, multiplied (annealed) by 0.1 at epochs 35, 50, 65, and 80.

ResNet20 (code modified from Idelbayev (2020)) were trained for 200 epochs using a minibatch size of 256 samples and a learning rate of 0.1, annealed by 0.1 at epochs 100 and 150.

### A.1.2 DATASETS

Tiny Imagenet (Fei-Fei et al., 2015) consists of 500 training images and 50 images for each of its 200 classes. We used the validation set for testing and created a new validation set by taking 50 images per class from the training set, selected randomly for each seed. We split the 50k CIFAR10 training samples into a 45k sample training set and a 5k validation set, similar to our approach with Tiny Imagenet.

All experimental results were derived from the test set with the parameters from the epoch that achieved the highest validation set accuracy over the training epochs. 20 replicates with different random seeds were run for each hyperparameter set. Selectivity regularization was not applied to the final (output) layer, nor was the final layer included any of our analyses.

CIFAR10C consists of a dataset in which 19 different naturalistic corruptions have been applied to the CIFAR10 test set at 5 different levels of severity. Tiny ImageNetC also has 5 levels of corruption severity, but consists of 15 corruptions.

We would like to note that Tiny ImageNetC does not use the Tiny ImageNet test data. While the two datasets were created using the same data generation procedure—cropping and scaling images from the same 200 ImageNet classes—they differ in the specific ImageNet images they use. It is possible that the images used to create Tiny ImageNetC are out-of-distribution with regards to the Tiny ImageNet training data, in which case our results from testing on Tiny ImageNetC actually underestimate the corruption robustness of our networks. The creators of Tiny ImageNetC kindly provided the clean (uncorrupted) Tiny ImageNetC data necessary for the dimensionality analysis, which relies on matches corrupted and clean data samples.

### A.1.3 SOFTWARE

Experiments were conducted using PyTorch (Paszke et al., 2019), analyzed using the SciPy ecosystem (Virtanen et al., 2019), and visualized using Seaborn (Waskom et al., 2017).

### A.1.4 QUANTIFYING DIMENSIONALITY

We quantified the dimensionality of a layer's representations by applying PCA to the layer's activation matrix for the clean test data and counting the number of dimensions necessary to explain 95% of the variance, then dividing by the total number of dimensions (i.e. the fraction of total dimensionality; we also replicated our results using the fraction of total dimensionality necessary to explain 90% and 99% of the variance). The same procedure was applied to compute the dimensionality of perturbation-induced changes in representations, except the activations for a perturbed data set were subtracted

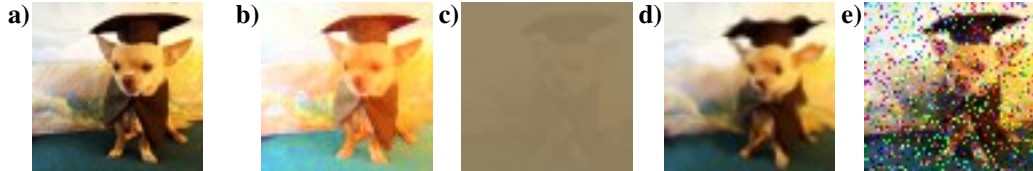

**Figure A1: Example naturalistic corruptions from the Tiny ImageNetC dataset.** (**a**) Clean (no corruption). (**b**) Brightness. (**c**) Contrast. (**d**) Elastic transform. (**e**) Shot noise. All corruptions are shown at severity level 5/5.

from the corresponding clean activations prior to applying PCA. For average-case perturbations, we performed this analysis for every corruption type and severity, and for the worst-case perturbations we used PGD with 40 steps.

Hidden layer representations in DNNs are known to lie on non-linear manifolds that are of lower dimensionality than the space in which they're embedded (Goodfellow et al., 2016; Ansuini et al., 2019). Consequently, linear methods such as PCA can fail to capture the "intrinsic" dimensionality of hidden layer representations. Thus we also quantified the intrinsic dimensionality (ID) of each layer's representations using the method of (Facco et al., 2017). The method, based on that of Levina and Bickel (2005), estimates ID by computing the ratio between the distances to the second and first nearest neighbors of each data point. We used the implementation of Ansuini et al. (2019). Our procedure was otherwise identical as when computing the linear dimensionality: we computed the dimensionality across all test data for each layer, then divided by the number of units per layer. We then computed the dimensionality of perturbation-induced changes in representations, except the activations for a perturbed data set were subtracted from the corresponding clean activations prior to computed ID. For average-case perturbations, we performed this analysis for every corruption type and severity, and for the worst-case perturbations we used PGD with 40 steps.

## A.2 EFFECTS OF CLASS SELECTIVITY REGULARIZATION ON TEST ACCURACY

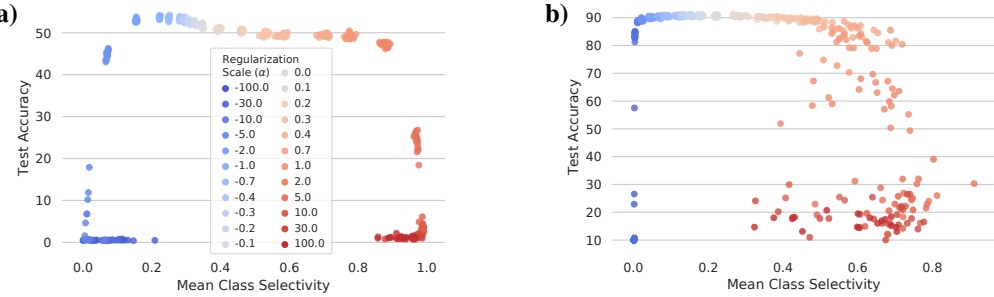

**Figure A2: Effects of class selectivity regularization on test accuracy.** Replicated as in Leavitt and Morcos (2020). (**a**) Test accuracy (y-axis) as a function of mean class selectivity (x-axis) for ResNet18 trained on Tiny ImageNet. $\alpha$ denotes the sign and intensity of class selectivity regularization. Negative $\alpha$ lowers selectivity, positive $\alpha$ increases selectivity, and the magnitude of $\alpha$ changes the strength of the effect. Each data point represents the mean class selectivity across all units in a single trained model. (**b**) Same as (**a**), but for ResNet20 trained on CIFAR10.

### A.3 ADDITIONAL RESULTS FOR AVERAGE-CASE PERTURBATION ROBUSTNESS

Because modifying class selectivity can affect performance on clean (unperturbed) inputs (Leavitt and Morcos (2020); Figure A2), it is possible that the effects we observe of class selectivity on perturbed test accuracy are not caused by changes in perturbation robustness per se, but simply by changes in baseline model accuracy. We controlled for this by normalizing each model's perturbed test accuracy by its clean (unperturbed) test accuracy. The results are generally consistent even after controlling for clean test accuracy, although increasing class selectivity does not cause the same deficits in as measured using non-normalized perturbed test accuracy in ResNet18 trained on Tiny ImageNet (Figure A3a). Interestingly, in ResNet20 trained on CIFAR10, normalizing perturbed test accuracy reveals a more dramatic improvement in perturbation robustness caused by reducing class selectivity (Figure A6c). The results for Resnet50 trained on Tiny ImageNet are entirely consistent between raw vs. normalized measures (Figure A9b vs. Figure A9c)

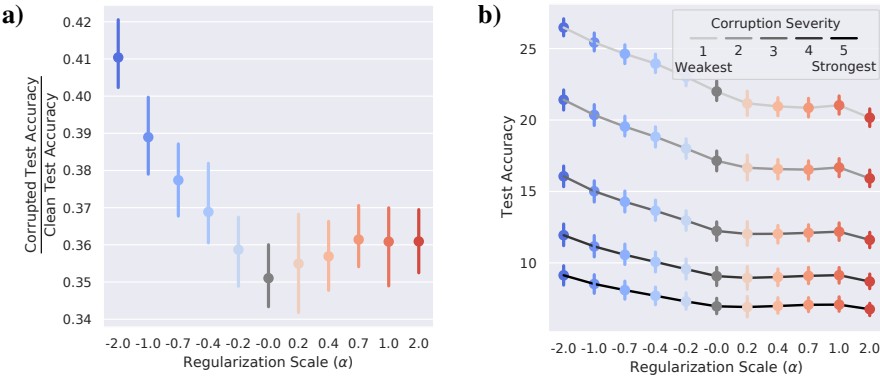

**Figure A3: Controlling for clean test accuracy, and effect of corruption severity across corruptions.** (**a**) Corrupted test accuracy normalized by clean test accuracy (y-axis) as a function of class selectivity regularization scale ($\alpha$; x-axis). Negative $\alpha$ lowers selectivity, positive $\alpha$ increases selectivity, and the magnitude of $\alpha$ changes the strength of the effect. Normalized perturbed test accuracy appears higher in networks with high class selectivity (large $\alpha$), but this is likely due to a floor effect: clean test accuracy is already much closer to the lower bound—chance—in networks with very high class selectivity, which may reflect a different performance regime, making direct comparison difficult. (**b**) Mean test accuracy across all corruptions (y-axis) as a function of $\alpha$ (x-axis) for different corruption severities (ordering along y-axis; shade of connecting line). Error bars indicate 95% confidence intervals of the mean.

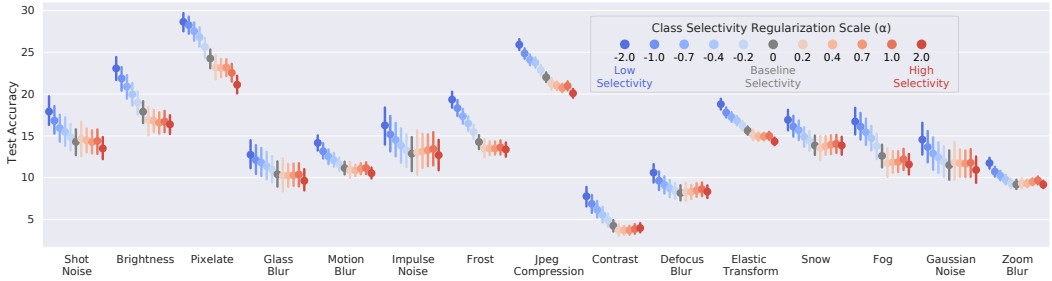

**Figure A4: Mean test accuracy across corruption intensities for each corruption type for ResNet18 tested on Tiny ImageNetC.** Test accuracy (y-axis) as a function of corruption type (x-axis) and class selectivity regularization scale ($\alpha$, color). Reducing class selectivity improves robustness against all 15/15 corruption types.

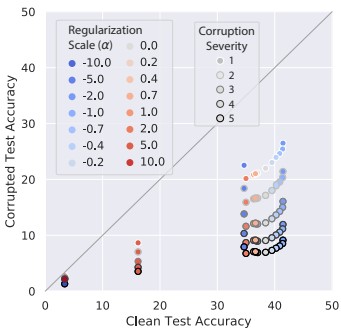

**Figure A5: Trade-off between clean and perturbed test accuracy in ResNet18 tested on Tiny ImageNetC.**
Clean test accuracy (x-axis) vs. perturbed test accuracy (y-axis) for different corruption severities (border color) and regularization scales ($\alpha$, fill color). Mean is computed across all corruption types. Error bars = 95% confidence intervals of the mean.

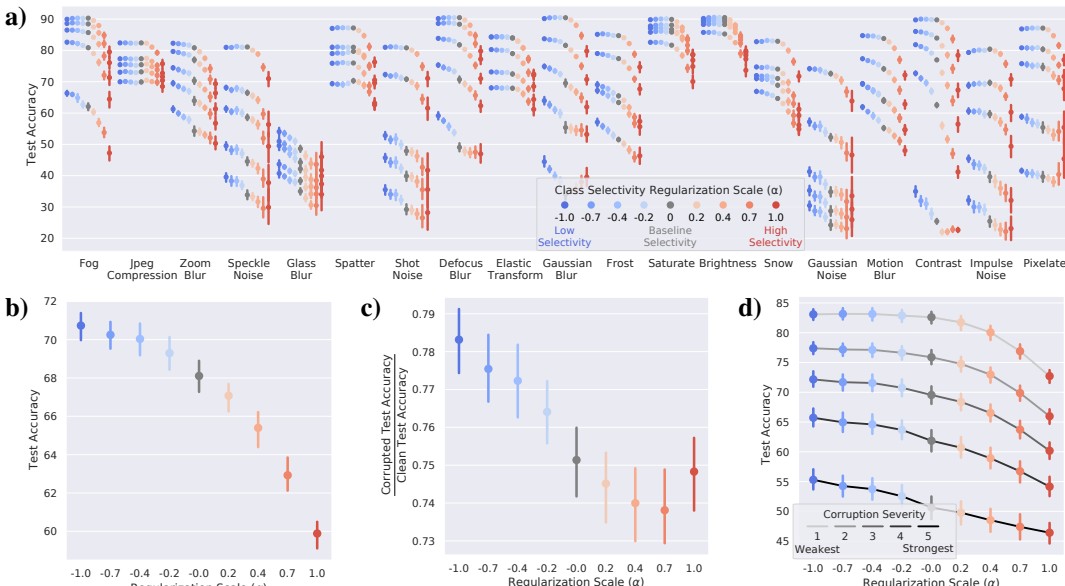

**Figure A6: Reducing class selectivity confers robustness to average-case perturbations in ResNet20 tested on CIFAR10C.** (**a**) Test accuracy (y-axis) as a function of corruption type (x-axis), class selectivity regularization scale ($\alpha$; color), and corruption severity (ordering along y-axis). Test accuracy is reduced proportionally to corruption severity, leading to an ordering along the y-axis, with corruption severity 1 (least severe) at the top and corruption severity 5 (most severe) at the bottom. Negative $\alpha$ lowers selectivity, positive $\alpha$ increases selectivity, and the magnitude of $\alpha$ changes the strength of the effect (see Figure A2b and Approach 3). (**b**) Mean test accuracy across all corruptions and severities (y-axis) as a function of $\alpha$ (x-axis). (**c**) Corrupted test accuracy normalized by clean test accuracy (y-axis) as a function of class selectivity regularization scale ($\alpha$; x-axis). Negative $\alpha$ lowers selectivity, positive $\alpha$ increases selectivity, and the magnitude of $\alpha$ changes the strength of the effect. Normalized perturbed test accuracy appears higher in networks with higher class selectivity (larger $\alpha$), but this is likely due to a floor effect: clean test accuracy is already much closer to the lower bound—chance—in networks with very high class selectivity, which may reflect a different performance regime, making direct comparison difficult. (**d**) Mean test accuracy across all corruptions (y-axis) as a function of $\alpha$ (x-axis) for different corruption severities (ordering along y-axis; shade of connecting line). Error bars indicate 95% confidence intervals of the mean.

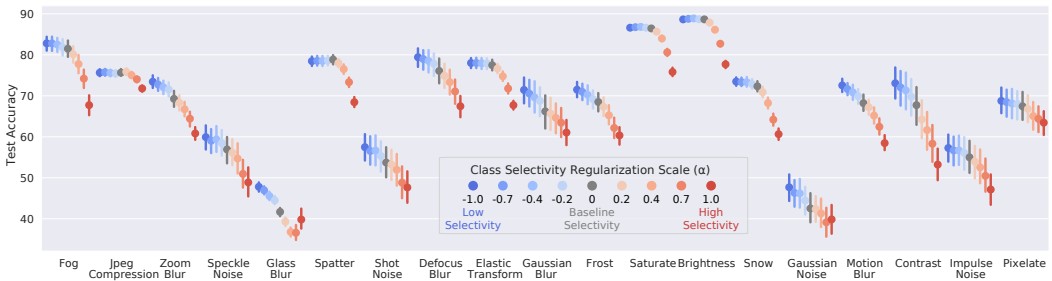

**Figure A7: Mean test accuracy across corruption intensities for each corruption type for ResNet20 tested on CIFAR10C.** Test accuracy (y-axis) as a function of corruption type (x-axis) and class selectivity regularization scale ($\alpha$, color). Reducing class selectivity improves robustness against 14/19 corruption types. Error bars = 95% confidence intervals of the mean.

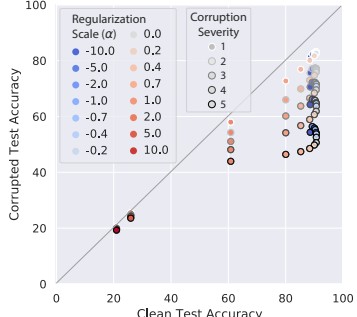

**Figure A8: Trade-off between clean and corrupted test accuracy in ResNet20 tested on CIFAR10C.** Clean test accuracy (x-axis) vs. corrupted test accuracy (y-axis) for different corruption severities (border color) and regularization scales ($\alpha$, fill color). Mean is computed across all corruption types.

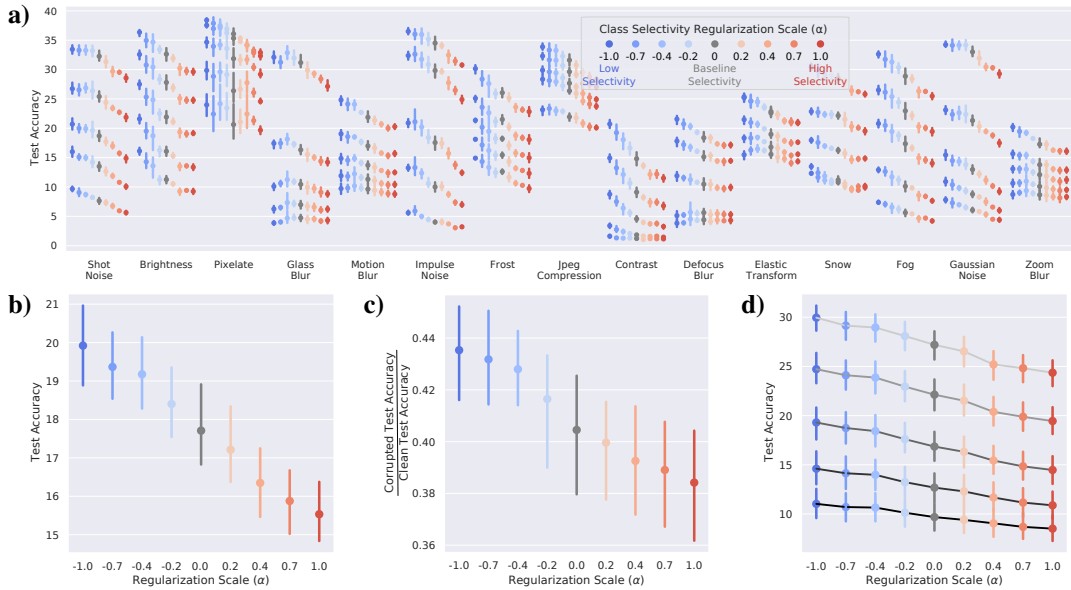

**Figure A9: Reducing class selectivity confers robustness to average-case perturbations in ResNet50 tested on Tiny ImageNetC.** (**a**) Test accuracy (y-axis) as a function of corruption type (x-axis), class selectivity regularization scale ($\alpha$; color), and corruption severity (ordering along y-axis). Test accuracy is reduced proportionally to corruption severity, leading to an ordering along the y-axis, with corruption severity 1 (least severe) at the top and corruption severity 5 (most severe) at the bottom. Negative $\alpha$ lowers selectivity, positive $\alpha$ increases selectivity, and the magnitude of $\alpha$ changes the strength of the effect (see Approach 3). (**b**) Mean test accuracy across all corruptions and severities (y-axis) as a function of $\alpha$ (x-axis). (**c**) Corrupted test accuracy normalized by clean test accuracy (y-axis) as a function of class selectivity regularization scale ($\alpha$; x-axis). Negative $\alpha$ lowers selectivity, positive $\alpha$ increases selectivity, and the magnitude of $\alpha$ changes the strength of the effect. (**d**) Mean test accuracy across all corruptions (y-axis) as a function of $\alpha$ (x-axis) for different corruption severities (ordering along y-axis; shade of connecting line). Error bars indicate 95% confidence intervals of the mean. Note that confidence intervals are larger in part due to a smaller sample size—only 5 replicates per $\alpha$ instead of 20.

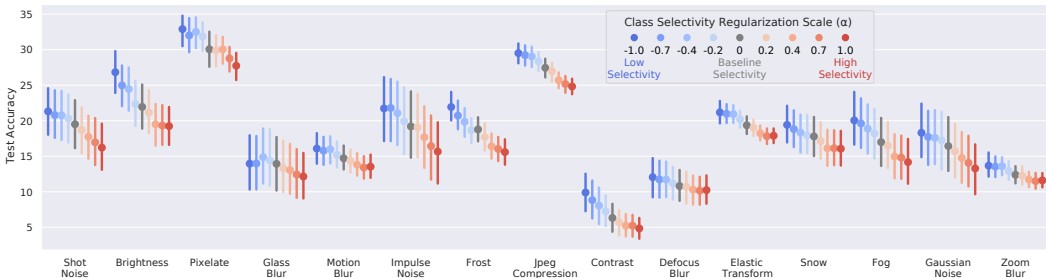

**Figure A10: Mean test accuracy across corruption intensities for each corruption type for ResNet50 tested on Tiny ImageNetC.** Test accuracy (y-axis) as a function of corruption type (x-axis) and class selectivity regularization scale ($\alpha$, color). Reducing class selectivity improves robustness against 14/15 corruption types. Error bars = 95% confidence intervals of the mean. Note that confidence intervals are larger in part due to a smaller sample size—only 5 replicates per $\alpha$ instead of 20.

### A.4 THE CAUSAL RELATIONSHIP BETWEEN CLASS SELECTIVITY AND AVERAGE-CASE ROBUSTNESS IS BIDIRECTIONAL

We found that regularizing to decrease class selectivity causes robustness to average-case perturbations. But is the converse is also true? Does increasing robustness to average-case perturbations also cause class selectivity to increase? We investigated this question by training with AugMix, a technique known to improve worst-case robustness (Hendrycks et al., 2020a). Briefly, AugMix stochastically applies a diverse set of image augmentations and uses a Jensen-Shannon Divergence consistency loss. Our AugMix parameters were as follows: mixture width: 3; mixture depth: stochastic; augmentation probability: 1; augmentation severity: 2. We found that AugMix does indeed decrese the mean level of class selectivity across neurons in a network (Figure A11). AugMix decreases overall levels of selectivity similarly to training with a class selectivity regularization scale of approximately $\alpha = -0.1$ or $\alpha = -0.2$ in both ResNet18 trained on Tiny ImageNet (Figures A11a and A11b) and ResNet20 trained on CIFAR10 (Figures A11c and A11d). These results indicate that the causal relationship between average-case perturbation robustness and class selectivity is bidirectional: not only does decreasing class selectivity cause average-case perturbation robustness to increase, but increasing average-case perturbation-robustness also causes class selectivity to decrease.

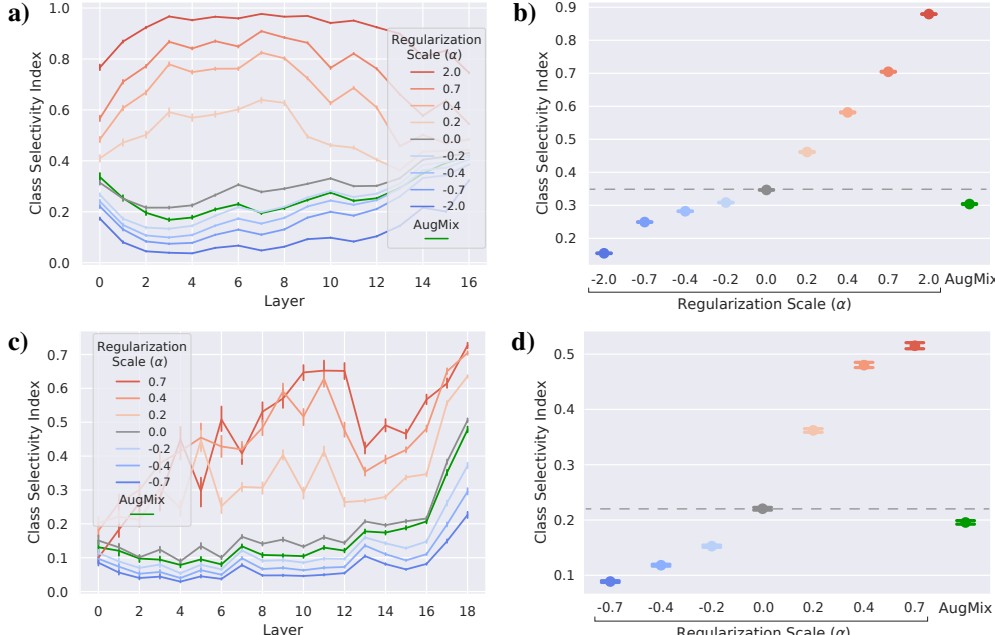

**Figure A11: AugMix training causes class selectivity to decrease.** (**a**) Mean class selectivity (y-axis) as a function of layer (x-axis) and class selectivity regularization scale ($\alpha$) or when training with AugMix (hue; AugMix in green) for ResNet18 trained on Tiny ImageNet. (**b**) Similar to (**a**), but mean class selectivity is computed across an entire network (y-axis) for different class selectivity regularization scales ($\alpha$) or when training with AugMix (x-axis; AugMix in green) for ResNet18 trained on Tiny ImageNet. The dashed line denotes the mean class selectivity for $\alpha = 0$, the baseline for comparison to AugMix. (**c**) and (**d**), identical to (**a**) and (**b**), but for ResNet20 trained on CIFAR10. Error bars denote 95% confidence intervals of the mean.

### A.5 WORST-CASE PERTURBATION ROBUSTNESS

We also confirmed that the worst-case robustness of high-selectivity ResNet18 and ResNet20 networks was not simply due to gradient-masking (Athalye et al., 2018) by generating worst-case perturbations using each of the replicate models trained with no selectivity regularization ($\alpha = 0$), then testing selectivity-regularized models on these samples. We found that high-selectivity models were less vulnerable to the $\alpha = 0$ samples than low-selectivity models for high-intensity perturbations (Appendix A14, indicating that gradient-masking does not fully account for the worst-case robustness of high-selectivity models.

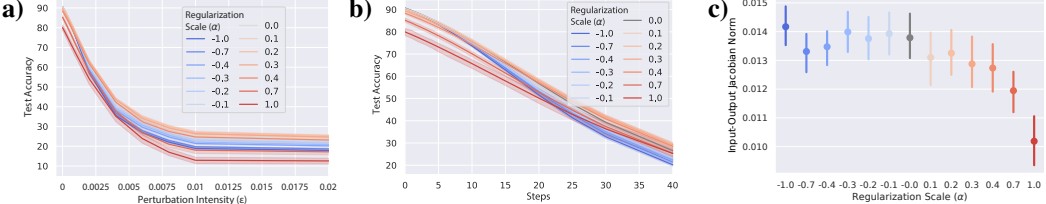

**Figure A12: Reducing class selectivity increases worst-case perturbation vulnerability in ResNet20 trained on CIFAR10.** (**a**) Test accuracy (y-axis) as a function of perturbation intensity ($\epsilon$; x-axis) and class selectivity regularization scale ($\alpha$; color) for the FGSM attack. (**b**) Test accuracy (y-axis) as a function of adversarial optimization steps (x-axis) and $\alpha$ (color) for the PGD attack. (**c**) Network stability, as measured with norm of the input-output Jacobian (y-axis) as a function of $\alpha$ (x-axis).

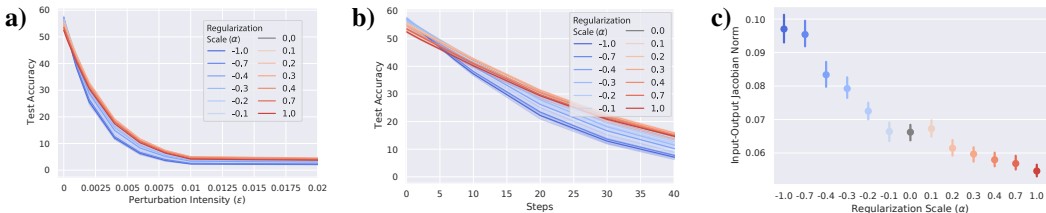

**Figure A13: Reducing class selectivity increases worst-case perturbation vulnerability in ResNet50 trained on Tiny ImageNet.** (**a**) Test accuracy (y-axis) as a function of perturbation intensity ($\epsilon$; x-axis) and class selectivity regularization scale ($\alpha$; color) for the FGSM attack. (**b**) Test accuracy (y-axis) as a function of adversarial optimization steps (x-axis) and $\alpha$ (color) for the PGD attack. (**c**) Network stability, as measured with norm of the input-output Jacobian (y-axis) as a function of $\alpha$ (x-axis).

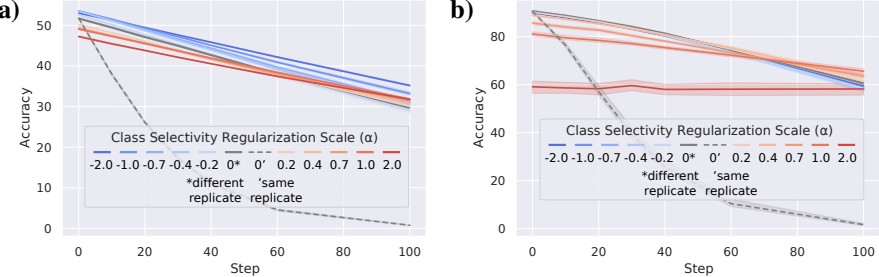

**Figure A14: Gradient masking does not fully account for worst-case perturbation robustness conferred by increased class selectivity.** (**a**) Test Accuracy (y-axis) as a function of adversarial optimization steps (x-axis) and class selectivity regularization scale ($\alpha$; color) when tested on adversarial examples generated using $\alpha = 0$. Because 20 replicate networks were trained for each value of $\alpha$ (see Approach 3), models trained with $alpha = 0$ could be tested on adversarial examples generated from a different replicate $\alpha = 0$ network ("different replicate"; solid line) or adversarial samples generated from their own parameters ("same replicate"; dashed line). Data shown are for ResNet18 trained on Tiny ImageNet (**b**). Same as (**a**), but for ResNet20 trained on CIFAR10.

### A.6 CLASS SELECTIVITY REGULARIZATION DOES NOT AFFECT ROBUSTNESS TO NATURAL ADVERSARIAL EXAMPLES

We also examined whether class selectivity regularization affects robustness to "natural" adversarial examples, images that are "natural, unmodified, real-world examples...selected to cause a fixed model to make a mistake" (Hendrycks et al., 2020b). We tested robustness to natural adversarial examples using ImageNet-A, a dataset of natural adversarial examples that belong to ImageNet classes but consistently cause misclassification errors with high confidence (Hendrycks et al., 2020b). We adapted ImageNet-A to our models trained on Tiny ImageNet (ResNet18 and ResNet50) by only testing on the 74 image classes that overlap between ImageNet-A and Tiny ImageNet (yielding a total of 2957 samples), and downsampling the images to 64 x 64. Test accuracy was similar across all tested values of $\alpha$ for both ResNet18 (Figure A15a) and ResNet50 (Figure A15b), indicating that class selectivity regularization may share some limitations with other methods for improving robustness, many of which also fail to yield significant robustness against ImageNet-A (Hendrycks et al., 2020b).

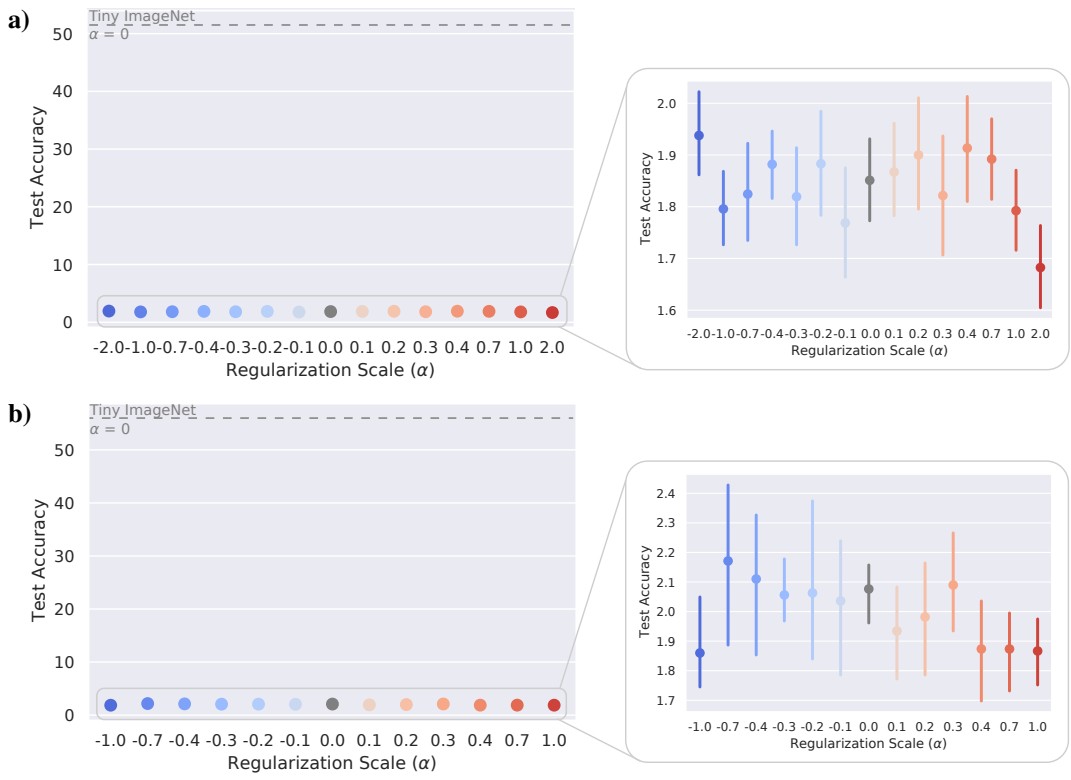

**Figure A15: Class selectivity regularization does not affect robustness to natural adversarial examples**
(**a**) Test accuracy on ImageNet-A (y-axis) as a function of regularization scale ($\alpha$; x-axis and hue) for ResNet18 trained on Tiny ImageNet. The dashed line at the top of the plot denotes the mean test accuracy on Tiny ImageNet for models trained with $\alpha = 0$, which serves as a baseline for comparison. (**b**) Identical to (**a**), but for ResNet50 trained on Tiny ImageNet. Error bars = 95% confidence interval of the mean. Note that confidence intervals are larger for ResNet50 in part due to a smaller sample size—only 5 replicates per $\alpha$ instead of 20. Chance = $\frac{1}{74}(1.35\%)$

## A.7 THE CAUSAL RELATIONSHIP BETWEEN CLASS SELECTIVITY AND WORST-CASE ROBUSTNESS IS BIDIRECTIONAL

We observed that regularizing to increase class selectivity causes robustness to worst-case perturbations. But is the converse is also true? Does increasing robustness to worst-case perturbations cause class selectivity to increase? We investigated this question using PGD training, a common technique for improving worst-case robustness. PGD training applies the PGD method of sample perturbation (see Approach 3) to samples during training. We used the same parameters for PGD sample generation when training our models as when testing (Approach 3). The number of PGD iterations controls the intensity of the perturbation, and the degree of perturbation-robustness in the trained model (Madry et al., 2018). We found that PGD training does indeed increase the mean level of class selectivity across neurons in a network, and this effect is proportional to the strength of PGD training: networks trained with more strongly-perturbed samples have higher class selectivity (Figure A16). Interestingly, PGD training also appear to cause units to die (Lu et al., 2019), and the number of dead untis is proportional to the intensity of PGD training (Figures A16b and A16e). Removing dead units, which have a class selectivity index of 0, from the calculation of mean class selectivity results in a clear, monotonic effect of PGD training intensity on class selectivity in both ResNet18 trained on Tiny ImageNet (Figure A16c) and ResNet20 trained on CIFAR10 (Figure A16f). These results indicate that the causal relationship between worst-case perturbation robustness and class selectivity is bidirectional: increasing class selectivity not only causes increased worst-case perturbation robustness, but increasing worst-case perturbation-robustness also causes increased class selectivity.

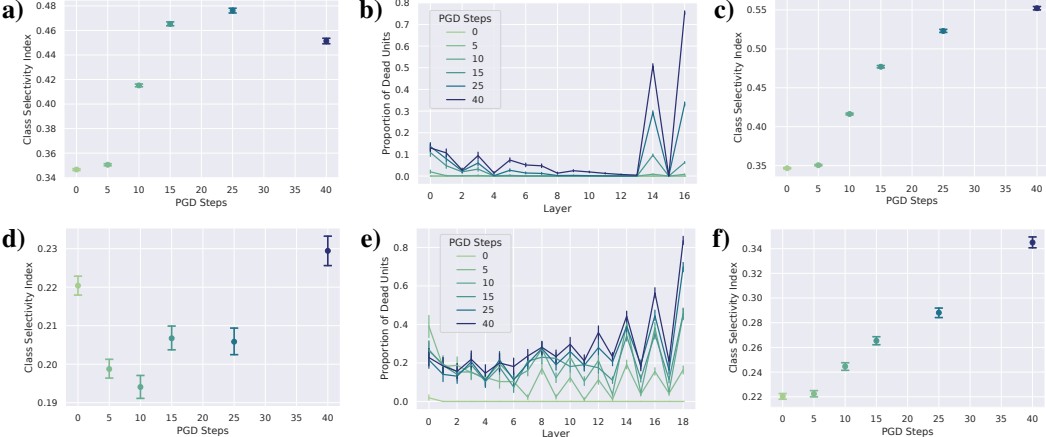

**Figure A16: PGD training causes class selectivity to increase.** (a) Mean class selectivity (y-axis) as a function of PGD training steps (x-axis; hue) for ResNet18 trained on Tiny ImageNet. The number of PGD steps corresponds to the intensity of PGD training and worst-case robustness. (b) Proportion of dead units (y-axis) as a function of layer (x-axis) for different numbers of PGD training steps (hue). (c) Mean class selectivity (y-axis) as a function of PGD training steps (x-axis; hue) after removing dead units from calculation of mean. (d) - (f), identical to (a) - (c), but for ResNet20 trained on CIFAR10.

A.8 STABILITY TO INPUT PERTURBATIONS IN UNITS AND LAYERS

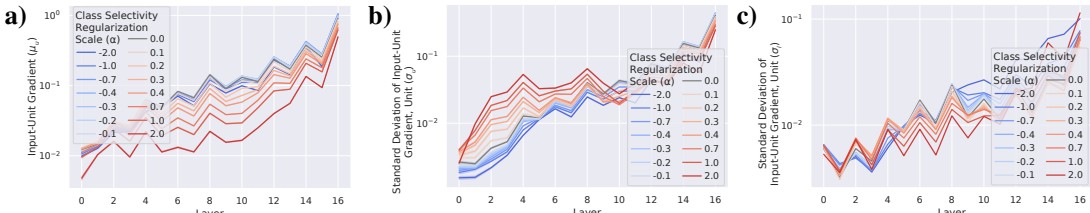

**Figure A17: Mean and standard deviation of input-unit gradient in ResNet18 trained on Tiny ImageNet.** (**a**) Input-unit gradient (y-axis) as a function of layer (x-axis). (**b**) Standard deviation of input-unit gradient for each unit ($\sigma_u$; y-axis) as a function of layer (x-axis). (**c**) Standard deviation of input-unit gradient across units in a layer ($\sigma_l$; y-axis) as a function of layer (x-axis). Shaded region = 95% confidence interval of the mean.

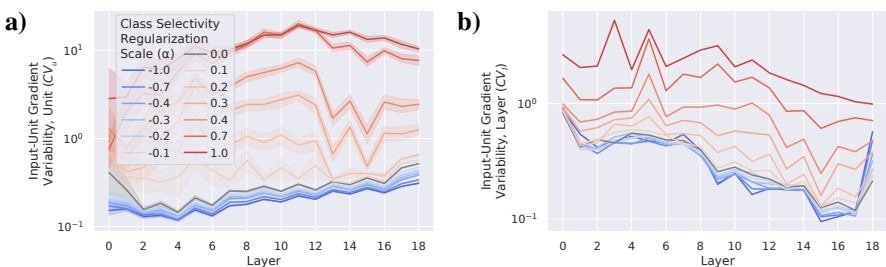

**Figure A18: Class selectivity causes higher variation in perturbability within and across units in ResNet20 trained on CIFAR10.** (**a**) Coefficient of variation of input-unit gradient for each unit ($CV_u$; see Approach 3; y-axis) as a function of layer (x-axis). (**b**) CV of input-unit gradient across units in a layer ($CV_l$; y-axis) as a function of layer (x-axis). Shaded regions = 95% confidence interval of the mean.

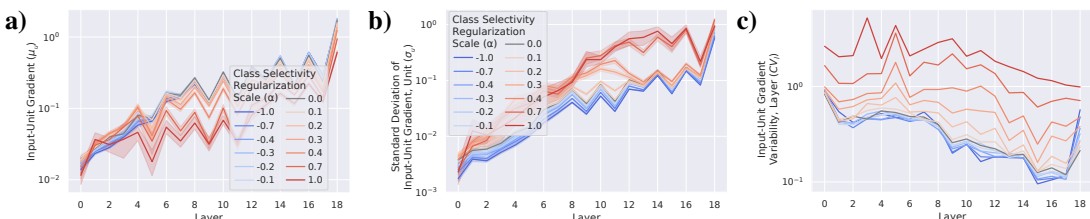

**Figure A19: Mean and standard deviation of input-unit gradient in ResNet20 trained on CIFAR10.** (**a**) Input-unit gradient (y-axis) as a function of layer (x-axis). (**b**) Standard deviation of input-unit gradient for each unit ($\sigma_u$; y-axis) as a function of layer (x-axis). (**c**) Standard deviation of input-unit gradient across units in a layer ($\sigma_l$; y-axis) as a function of layer (x-axis). Shaded region = 95% confidence interval of the mean.

### A.9 REPRESENTATIONAL DIMENSIONALITY

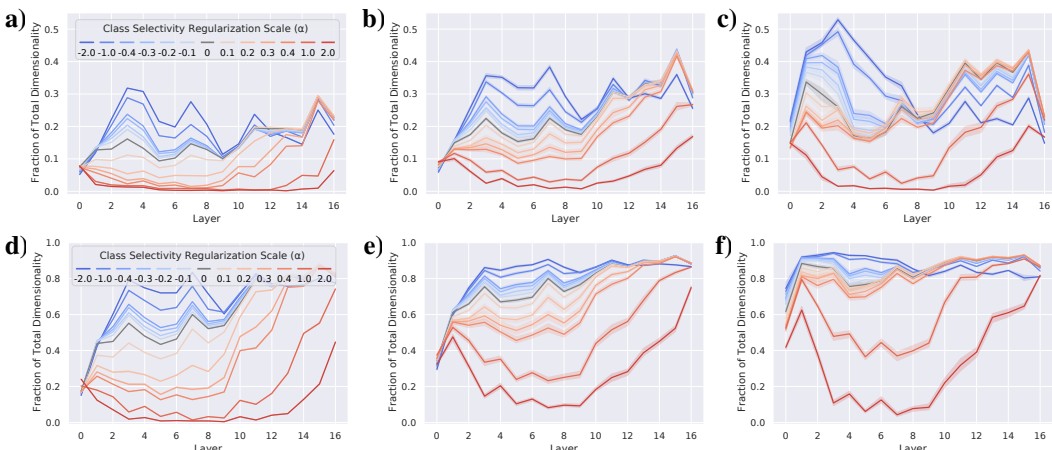

**Figure A20: Dimensionality in early layers predicts worst-case vulnerability in ResNet18 trained on Tiny ImageNet.** Identical to Figure 4, but dimensionality is computed as the number of principal components needed to explain 90% of variance in **(a)** - **(c)**, and 99% of variance in **(d)** - **(f)**. **(a)** Fraction of dimensionality (y-axis; see Appendix A.1.4) as a function of layer (x-axis). **(b)** Dimensionality of difference between clean and average-case perturbation activations (y-axis) as a function of layer (x-axis). **(c)** Dimensionality of difference between clean and worst-case perturbation activations (y-axis) as a function of layer (x-axis). **(d)** - **(f)**, identical to **(a)** - **(c)**, but for 99% explained variance threshold.

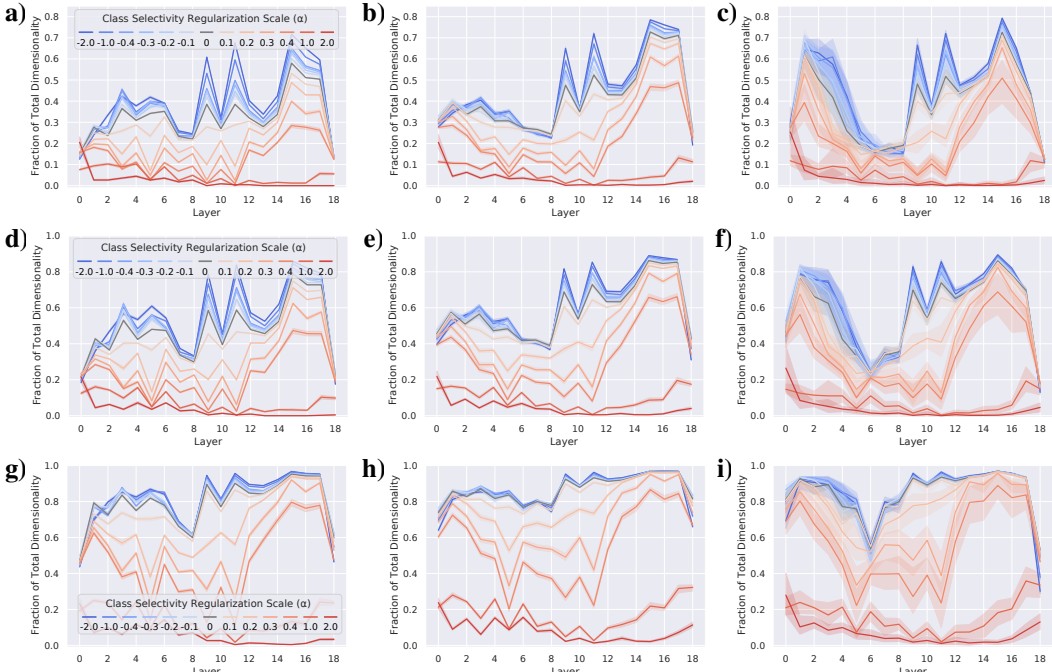

**Figure A21: Dimensionality in early layers predicts worst-case vulnerability in ResNet20 trained on CIFAR10.** Identical to Figure 4, but for ResNet20 trained on CIFAR10. **(a)** Fraction of dimensionality needed to explain 90% of variance (y-axis; see Appendix A.1.4) as a function of layer (x-axis). **(b)** Dimensionality of difference between clean and average-case perturbation activations (y-axis) as a function of layer (x-axis). **(c)** Dimensionality of difference between clean and worst-case perturbation activations (y-axis) as a function of layer (x-axis). **(d)** - **(f)**, identical to **(a)** - **(c)**, but for 95% explained variance threshold. **(g)** - **(i)**, identical to **(a)** - **(c)**, but for 99% explained variance threshold.

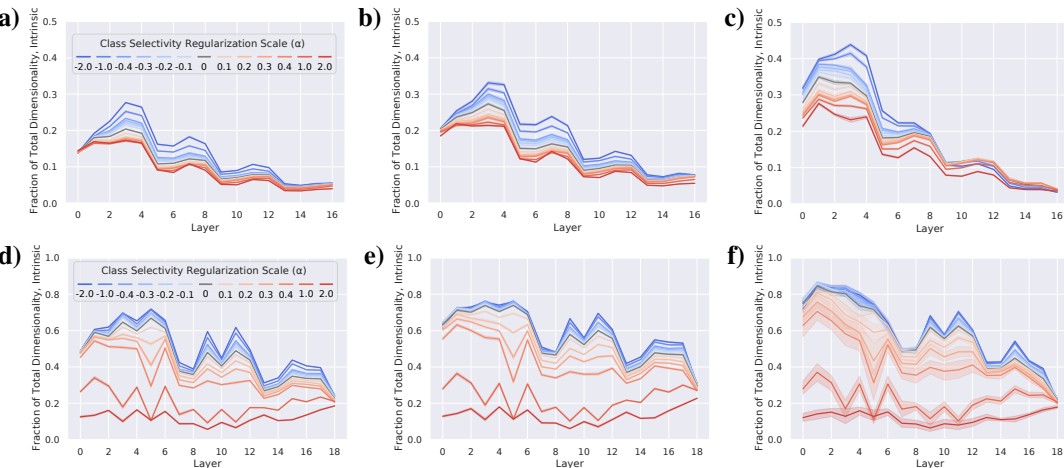

**Figure A22: Intrinsic dimensionality in early layers predicts worst-case vulnerability.** (**a**) Fraction of intrinsic dimensionality (y-axis; see Appendix A.1.4) as a function of layer (x-axis) for ResNet18 trained on Tiny ImageNet. (**b**) Dimensionality of difference between clean and average-case perturbation activations (y-axis) as a function of layer (x-axis) for ResNet18 trained on Tiny ImageNet. (**c**) Dimensionality of difference between clean and worst-case perturbation activations (y-axis) as a function of layer (x-axis) for ResNet18 trained on Tiny ImageNet. (**d**) - (**f**), identical to (**a**) - (**c**), but for ResNet20 trained on CIFAR10.

