# OpenReview forum: "Linking average- and worst-case perturbation robustness via class selectivity and dimensionality"
_ICLR.cc/2021/Conference — Reject_

### Official Review · AnonReviewer3 · 2020-10-25
**Interesting empirical findings on robustness tradeoff with limited scope**

**Rating:** 6
**Confidence:** 4

**Review:**

Summary:
This paper studies the relationship between class selectivity and robustness. In particular, it finds that higher class selectivity tends to lead to worse average-case robustness (a.k.a. performance on naturalistically perturbed inputs) but better worst-case robustness (a.k.a. performance on adversarial inputs).

It further finds that variability of input-unit gradient across samples and units is proportional to a network’s overall class selectivity. Besides, the increase of dimensionality of activation caused by corruption is larger for worst-case perturbation and low-selectivity networks.

################################################

Reasons for score:
Overall, this paper makes a series of interesting observations on the relationship between class selectivity and robustness. I feel that the paper’s contribution is a bit limited to some interesting empirical observations on a particular task (image classification) and type of model (resnet) on two datasets (CIFAR-10 and Tiny ImageNet). Also, no attempt to make much theoretical connections and the practical impact of the paper is not very clear.

################################################

Pros:

+mostly well-written and easy to follow

+interesting empirical findings and exploration of sensitivity on worst-case robustness and average-case robustness

+experiments support the claims

Cons:

-evaluation looks a bit limited in terms of tasks/models/datasets.

As even the authors stated in the paper “We hesitate to generalize too broadly about our findings, as they are limited to CNNs trained on image classification tasks. It is possible that the results we report here are specific to our models and/or datasets, and also may not extend to other tasks.” The results are indeed a bit limited in terms of its scope. The evaluation is limited to only ResNets (Res18 and Res20, which are similar to each other in terms of properties) for the image classification task only. I wonder if the findings can hold for deeper models since Fig 3 (b) seems to show that for deeper layers the variability of different SI tends to be increasingly similar.


-pure empirical results and not enough theoretical justification

I understand that this is the first paper trying to connect average-case robustness and worst-case robustness with class selectivity but I still would like to see a bit more discussion on the theoretical side.

-the impact of the observations need more discussions.

I understand that this paper focuses on empirically linking average-case robustness and worst-case robustness. However, I think it needs a bit more discussion on how other researchers/practitioners can benefit from these findings. For example, how should one leverage such observations to detect adversarial / naturally perturbed error-prone input , improve robustness, or find the optimal tradeoff between different measures (e.g. natural accuracy, average-case robustness, worst-case robustness.

################################################

Typo:
In Figure A6 caption “Figure ??”

################################################

Questions:
At the end of sec 4.2 “indicating that distributed semantic representations are more vulnerable to worst-case perturbation because they are less stable than sparse semantic representations.” What does “distributed semantic representations” mean here? What’s the difference between “distributed semantic representations” and “sparse semantic representations“ in this sentence?

################################################

Suggestions:
The paper can be improved by addressing at least one of the three cons I mentioned.


################################################

Post-Rebuttal:
Thanks the authors for their detailed responses!
The authors responses addressed most of my concerns. In particular,1. the authors showed that their results generalize to Tiny-ImageNet and ResNet50 with additional experiments. 2. the authors added some discussions on the theoretical side. Besides, I also appreciate the addition of experiments on AugMix and PGD which imply the bidirectional causality of  class selectivity and perturbation robustness. Overall, I think this work will potentially be a good addition to the existing understanding of trade-off between worst-case robustness and average-case robustness for the community. Therefore, I decide to increase my score to 6.

---

> ### Author Response · Authors · 2020-11-25
> **Response to reviewer 3**
>
> Thank you for our helpful feedback. We revised the paper as per your suggestions and believe it has yielded substantial improvements. Please note our response is split across two comments due to space limitations.
>
> >evaluation looks a bit limited in terms of tasks/models/datasets. As even the authors stated in the paper “We hesitate to generalize too broadly about our findings, as they are limited to CNNs trained on image classification tasks. It is possible that the results we report here are specific to our models and/or datasets, and also may not extend to other tasks.” The results are indeed a bit limited in terms of its scope. The evaluation is limited to only ResNets (Res18 and Res20, which are similar to each other in terms of properties) for the image classification task only.
>
> This is a fair concern. We focus on ResNets because they are the standard of choice for image-related deep learning research, as prior models like InceptionNet, VGG, etc. have increasingly fallen out of use. We also understand the concern about the results being limited to image classification tasks, but the adversarial example literature and robustness literature primarily addresses image classification, so we would argue that our scope is consistent with what has generally been considered in prior work.
>
> However, in an attempt to address your concern and expand the scope of our work and the strength of our conclusions, we also re-ran our core experiments on ResNet50 trained on Tiny ImageNet. We found that effects of selectivity on robustness to average-case and worst-case perturbations are completely consistent with what was observed in our other two models: Class selectivity is inversely proportional to average-case robustness (Figure A9) and confers improvements to 14 out of 15 corruptions in Tiny ImageNetC (Figures A9-10); and worst-case robustness and network stability are both proportional to class selectivity (Figure A13).

---

> > ### Author Response · Authors · 2020-11-25
> > **Response to reviewer 3, cont'd**
> >
> > >pure empirical results and not enough theoretical justification. I understand that this is the first paper trying to connect average-case robustness and worst-case robustness with class selectivity but I still would like to see a bit more discussion on the theoretical side…The impact of the observations need more discussions. I understand that this paper focuses on empirically linking average-case robustness and worst-case robustness. However, I think it needs a bit more discussion on how other researchers/practitioners can benefit from these findings. For example, how should one leverage such observations to detect adversarial / naturally perturbed error-prone input , improve robustness, or find the optimal tradeoff between different measures (e.g. natural accuracy, average-case robustness, worst-case robustness.
> >
> > Thank you for the suggestions regarding the importance of theoretical and practical discussion. We have added a segment discussing the relationship between class selectivity, sparsity, and dimensionality (Introduction, paragraph 3; Footnote 2; also see our response to your final question), and we have also updated the discussion to address both these issues. What follows is the updated portion of the discussion:
> >
> > We hesitate to generalize too broadly about our findings, as they are limited to CNNs trained on image classification tasks. It is possible that the results we report here are specific to our models and/or datasets, and also may not extend to other tasks. Scaling class selectivity regularization to datasets with large numbers of classes also remains an open problem (Leavitt and Morcos, 2020).
> >
> > Our findings could be utilized for practical ends and to clarify findings in prior work. Relevant to both of these issues is the task of adversarial example detection. There is conflicting evidence that intrinsic dimensionality can be used to characterize or detect adversarial (worst-case) samples (Ma et al., 2018; Lu et al., 2018). Our finding that worst-case perturbations cause a marked increase in both intrinsic and linear dimensionality indicates that there may be merit in continuing to study these quantities for use in worst-case perturbation detection.
> >
> > Our observation that the causal relationship between class-selectivity and robustness is bidirectional—increased class selectivity causes worst-case robustness, and vice-versa—helps clarify the known benefits of sparsity (Madry et al., 2018; Balda et al., 2020; Ye et al., 2018; Guo et al., 2018; Dhillon et al., 2018) and dimensionality (Langeberg et al., 2019; Sanyal et al., 2020; Nayebi and Ganguli, 2017) on worst-case robustness. It furthermore raises the question of whether the robustness benefits of enforcing low-dimensional representations also cause class selectivity to increase.
> >
> > Our work may also hold practical relevance to developing robust models: class selectivity could be used as both a metric for measuring model robustness and a method for achieving robustness (via regularization). We hope future work will more comprehensively assess the utility of class selectivity in the deep learning toolkit for this purpose.
> >
> > >Typo: In Figure A6 caption “Figure ??”
> >
> > Keen eyes! Thank you, fixed.
> >
> > >At the end of sec 4.2 “indicating that distributed semantic representations are more vulnerable to worst-case perturbation because they are less stable than sparse semantic representations.” What does “distributed semantic representations” mean here? What’s the difference between “distributed semantic representations” and “sparse semantic representations“ in this sentence?
> >
> > We apologize for the lack of clarity. Class selectivity in individual neurons measures how strongly class information, which is a semantic concept, is represented by that neuron.  Thus class selectivity measures the degree to which semantic concepts are represented in individual neurons, which could be considered a form of sparsity. A network with low levels of class selectivity has distributed semantic representations, while a network with high levels of class selectivity would have sparse semantic representations. For example, if a network has high test accuracy on a classification task, it is necessarily representing class (semantic) information. But if the mean class selectivity across units is low, then the individual units do not contain much class information, thus the class information must be distributed across units; the semantic representation in this case is not sparse, it is distributed.  We hope this explanation clarifies the issue for you. We have updated the text accordingly (Introduction, paragraph 3; Footnote 2).

---

### Official Review · AnonReviewer4 · 2020-10-28
**Results appear preliminary**

**Rating:** 4
**Confidence:** 3

**Review:**

##########################################################################

Summary:

This work empirically studies the relationship between robustness and class selectivity, a measure of neuron variability between classes. Robustness to both adversarial ("worst-case") perturbations and corruptions ("average-case") are considered. This work builds off the recent work of Leavitt and Morcos (2020) (currently in review at ICLR 2021) who claim empirical evidence that class selectivity may be harmful for generalization.  The experiments in this paper examine the robustness (in both senses) of networks explicitly regularized for class selectivity. The main empirical claims are that (1) class sensitivity is negatively correlated with robustness to corruptions (2) class sensitivity is positively correlated with robustness to adversarial perturbations.



##########################################################################

Reasons for score:


Overall I vote for rejection. The authors frame the results in connection to sparsity and dimensionality. But at present the evidence for this connection appears preliminary. For example the differences in the class selectivity curves in Figure 2 appear marginal. On the other hand, Figure 1 does seem convincing for the claim reducing class selectivity improves robustness to corruptions.


##########################################################################

Pros:
* Clear results in Figure 1a
* Important topic
* Potentially relevant results

##########################################################################

Cons:
* Marginal results in Figure 2
* Unclear results in Figure 4
* Measurements of dimensionality limited to PCA
* Not easy to read. The paper appears hastily written.

##########################################################################

Questions during rebuttal period:

Please argue why the differences between class-selective curves in Figure 2 are significant.

I am quite unclear on how to interpret Figure 4. Please clarify.

The authors use the word "causal" several times in the paper, which appears to me dubious. I can find no justification for a claim of causality here, since the results are correlative. Can the authors clarify this?

The authors only examine one dimensionality estimation method: the number of PCA components required to capture 95% of the data variance. Dimensionality estimation with PCA on data with non-linear (i.e. manifold) structure is problematic. Thus I have doubts on measurements of dimensionality used here. The authors may consider adding the method of Levina and Bickel [0]

[0] Maximum Likelihood Estimation of Intrinsic Dimension - Levina and Bickel (Neurips 2004)
https://papers.nips.cc/paper/2577-maximum-likelihood-estimation-of-intrinsic-dimension

#########################################################################

Additional Feedback:

It is the prerogative of the authors to choose the words they believe best express their message. However I lament the author's choice to use "worst-case perturbation" and "average-case perturbation" to refer to "adversarial attack"and "corruption". The literature on adversarial attacks is quite large at this point, and "adversarial" is the de facto terminology. The authors claim their terminology is more general, however I cannot see the justification given the widespread existing usage.

#########################################################################

POST-REBUTTAL RESPONSE:

I read the author's rebuttal but have decided to not increase my score. I still have doubts over the claims in this paper.

---

> ### Author Response · Authors · 2020-11-25
> **Response to reviewer 4**
>
> We appreciate your thoughtful feedback. Your requests for clarity and explanation have improved the readability of our work, and the intrinsic dimensionality analysis you recommended strengthens our conclusions. Please note our response is split over three comments due to space limitations.
>
> >Please argue why the differences between class-selective curves in Figure 2 are significant.
>
> Certainly! The clear ordering of curves by class-selectivity in Figure 2a and 2b demonstrates that worst-case robustness is directly proportional to class selectivity; networks with higher class selectivity (red) have shallower curves than networks with lower class selectivity (blue). The confidence intervals, denoted by the shaded region around each line, are in some cases so small that they appear non-existent, speaking to the significance of this effect. Precise quantities in addition to the plot may also help convey the magnitude effect. For Figure 2a, at $\epsilon$ = 0.002, test accuracy = {28.9, 27.2; 22.9) for alpha = {2.0, 0, -2.0}; at $\epsilon$ = 0.004, test accuracy = {17.0, 13.6, 8.8} for alpha = {2.0, 0, -2.0}; and at $\epsilon$ = 0.006, test accuracy = {10.1, 7.0, 3.4} for alpha = {2.0, 0, -2.0}.  We would also like to point out that the initial ordering of test accuracy by alpha at x=0 is the result of the baseline effect of class selectivity regularization on test accuracy in the absence of worst-case perturbations: test accuracy is anti-correlated with class selectivity. This highlights that the relationship between worst-case robustness and class selectivity is strong enough that it rapidly overcomes the initial bias at x=0. We have updated the manuscript to further clarify this (Results 4.2, paragraph 3).
>
> >I am quite unclear on how to interpret Figure 4. Please clarify.
>
> We apologize for the lack of clarity. We hope the following explanation is not didactic, but in the interest of comprehension we will provide a high level of detail. Figure 4a shows dimensionality per unit (y-axis) as a function of layer (x-axis) for different levels of class selectivity regularization (hue). The ordering of dimensionality by class selectivity demonstrates that representational dimensionality is inversely correlated with class selectivity. Like Figure 4a, Figure 4b plots dimensionality as a function of layer for models with different levels of class selectivity. But the precise quantity plotted in Figure 4b is the difference between the activations for clean test samples and Tiny ImageNet-C (i.e. average-case perturbed) test samples. This quantifies the dimensionality of the _change_ in the representation induced by the perturbation.  A small value indicates that the perturbation impacts fewer units, while a larger change indicates that more units are impacted. A value of zero indicates that the perturbation induces no change. Figure 4c is similar to Figure 4b, but for adversarial samples instead of Tiny ImageNet-C. The greater stratification between networks with low class selectivity in Figures 4b and 4c compared to Figure 4a indicates that changes in input to are more distributed (i.e. affect more units) in models with lowerer class selectivity. And the larger change in early-layer dimensionality in Figure 4c compared to Figure 4b indicates that the changes in dimensionality caused by worst-case perturbations are larger than the changes in dimensionality caused by average-case perturbations.
>
> In summary, there are three conclusions: 1) Representational dimensionality is inversely proportional to class selectivity; 2) the dimensionality of perturbation-induced changes is inversely proportional to class selectivity; 3) the dimensionality of perturbation-induced changes is greater for worst-case than for average-case perturbations. We hope this clarifies Figure 4!

---

> > ### Author Response · Authors · 2020-11-25
> > **Response to reviewer 4, Part 2**
> >
> > >The authors use the word "causal" several times in the paper, which appears to me dubious. I can find no justification for a claim of causality here, since the results are correlative. Can the authors clarify this?
> >
> > We regret that our use of causal claims in the text did not appear sufficiently justified. Previous studies examining the role of class selectivity have typically used single unit ablation, or measured the relationship between class selectivity and another variable of interest (e.g. test accuracy) when manipulating a third variable (e.g. batch norm) [e.g. 1-5]. In contrast with previous work, the approach of [6], which we use here, controls class selectivity directly by regularizing for or against in the loss function during training. Accordingly, we used causal language when manipulating class selectivity directly and it is the only variable being manipulated. You are correct that our experiments do not address causal relationships between e.g. dimensionality and robustness, which is why we use phrasing that avoids causal claims about these relationships, such as “the relationship between dimensionality and robustness is _mediated_ by class selectivity”. However your concern is noted, and we have updated the Class Selectivity subsection of the Related Work (Section 2.2) to make explicit the utility of the class selectivity regularizer for addressing the causal effect of class selectivity in the context of prior work.
> >
> > >The authors only examine one dimensionality estimation method: the number of PCA components required to capture 95% of the data variance. Dimensionality estimation with PCA on data with non-linear (i.e. manifold) structure is problematic. Thus I have doubts on measurements of dimensionality used here. The authors may consider adding the method of Levina and Bickel [7]
> >
> > This is an excellent point. As per your recommendation, we quantified the intrinsic dimensionality (ID) of each layer's representations (Results 4.4, final paragraph; Appendix A.1.4; Figure A22). We used the method of [8], which is an extension of your recommended method [7]; briefly, it estimates ID by computing the ratio between the distances to the second and first nearest neighbors of each data point. It would not be unexpected if the ID told a very different story about the relationship between class selectivity, dimensionality, and the effect of perturbations. But, interestingly, the results were qualitatively similar to what we observed when examining linear dimensionality (Figure A22): ID is inversely proportional to class selectivity; the activation changes caused by both worst- and average-case perturbations are higher-dimensional than the representations of clean data; and the increase in early-layer dimensionality is larger for worst-case perturbations than average-case perturbations. This effect was present in both ResNet18 trained on Tiny ImageNet (Figure A22a-c) and ResNet20 trained on CIFAR10 (Figure A22d-f). Thus both linear and non-linear measures of dimensionality imply that representational dimensionality may present a trade-off between worst-case and average-case perturbation robustness.

---

> > > ### Author Response · Authors · 2020-11-25
> > > **Response to reviewer 4, Part 3**
> > >
> > > >It is the prerogative of the authors to choose the words they believe best express their message. However I lament the author's choice to use "worst-case perturbation" and "average-case perturbation" to refer to "adversarial attack"and "corruption". The literature on adversarial attacks is quite large at this point, and "adversarial" is the de facto terminology. The authors claim their terminology is more general, however I cannot see the justification given the widespread existing usage.
> > >
> > > We wrestled with this issue when drafting the manuscript. We agree that the use of “adversarial” and “corruption” are more conventional than “worst-case” and “average-case”, which is why explicitly address our choice of terminology in the Introduction and expand on it in Footnote 1. However, our usage is not novel; we follow the example of Hendrycks and Dietterich [9] as well as [10,11]. In earlier drafts of the manuscript, we regularly used  “adversarial” and “corruption” in addition to “worst-case” and “average-case”, respectively, but we decided to make the change for two reasons: The first reason is consistency. We wanted to minimize the reader’s lexical overhead. The second reason is that it is our opinion that using the terms “adversarial” and “corruption” connotes a categorical difference between these two concepts, and this difference is not scientifically justified. And such a connotation runs counter to the central observations of our experiments, which demonstrate links between worst-case and average-case perturbations.
> > >
> > > [1] Morcos et al., On the importance of single directions for generalization, 2018
> > >
> > > [2] Amjad et al., Understanding Individual Neuron Importance Using Information Theory, 2018
> > >
> > > [3] Zhou et al., Revisiting the Importance of Individual Units in CNNs via Ablation, 2018
> > >
> > > [4] Donnelly and Roegiest, On Interpretability and Feature Representations: An Analysis of the Sentiment Neuron, 2019
> > >
> > > [5] Dalvi et al., What Is One Grain of Sand in the Desert? Analyzing Individual Neurons in Deep NLP Models, 2019
> > >
> > > [6] Leavitt and Morcos, Selectivity considered harmful: evaluating the causal impact of class selectivity in DNNs, 2020
> > >
> > > [7] Levina and Bickel, Maximum Likelihood Estimation of Intrinsic Dimension, 2004
> > >
> > > [8] Facco et al., Estimating the intrinsic dimension of datasets by a minimal neighborhood information, 2017
> > >
> > > [9] Hendrycks and Dietterich, Benchmarking Neural Network Robustness to Common Corruptions and Perturbations, 2019
> > >
> > > [10] Yin et al., A Fourier Perspective on Model Robustness in Computer Vision, 2019
> > >
> > > [11] Gilmer et al., Motivating the Rules of the Game for Adversarial Example Research, 2018

---

### Official Review · AnonReviewer2 · 2020-10-29
**Interesting Idea but More Experiments are Needed**

**Rating:** 7
**Confidence:** 4

**Review:**

This paper presents a new finding that class selectivity is negatively correlated with average-case corruptions (natural visual distortions) while positively correlated with worst-case corruptions (adversarial attacks). The authors then try to explain this phenomenon from two aspects: variability of input-unit gradient and dimensionality in early layers.

Pros:
1. Class selectivity is previously used as a metric to indicate model generalization or memorization. It is good to see the authors generalize its usage to measuring model robustness to natural corruptions and adversarial attacks.
2. The finding that robustness to average-case corruptions are negatively correlated with class selectivity is no surprise to me, since it has been previously shown that low class selectivity indicates better generalization ability [1].
The intriguing finding is that robustness to adversarial images is positively correlated with class selectivity. This is contour intuitive to me at first glance, but the explanations provided in Section 4.3 and 4.4 convince me.
3. If the conclusions in this paper holds, I think this is one step further from the well-know accuracy-robustness tradeoff [5,6]: not only do we have tradeoff between clean accuracy and adversarial accuracy [5,6], but also a (possibly inherent) tradeoff between average- and worst-case adversarial accuracy.

Cons:
The general idea to use class selectivity as an indicator for model robustness in this paper is very interesting.
But I do think more experiments and discussions are needed to explain these phenomenons, especially the different behaviors of average-and worst-case corruptions.
1. Adversarial training (e.g., [2]) can greatly improve adversarial robustness. If the conclusions in this paper holds, we would expect adversarially trained models to have much better class selectivity than normally trained models. I'm wondering whether the authors could kindly provide these results?
2. Similarly, there are some methods (e.g., AugMix [4]) improving model robustness to average-case corruptions. Does AugMix model has significantly lower class selectivity than normally trained models.
3. In my point of view, one possible explanation for the different behaviors between average-case and worst-case corruptions could be that normal adversarial images (generated by PGD, FGSM as in your paper) are out-of-distribution samples [3], while average-case corruptions are likely to be on-distribution. So I'm a bit curious about the behaviors of the "on–
manifold adversarial examples" defined in [3]: do they behave more like normal adversarial images (e.g., causing a larger increase in early-layer dimensionality) or more like average-case corruptions (causing a smaller increase in early-layer dimensionality )?

[1] On the importance of single directions for generalization. ICLR, 2018.

[2] Towards Deep Learning Models Resistant to Adversarial Attacks. ICLR, 2018.

[3] Disentangling Adversarial Robustness and Generalization. CVPR, 2019.

[4] AugMix: A Simple Data Processing Method to Improve Robustness and Uncertainty. ICLR, 2020.

[5] Robustness May Be at Odds with Accuracy. ICLR, 2019.

[6] Theoretically Principled Trade-off between Robustness and Accuracy. ICML, 2019.

Update:
Thanks the authors for their response. All my concerns are addressed and I decide to increase my score from 6 to 7.

---

> ### Author Response · Authors · 2020-11-25
> **Response to Reviewer 2**
>
> Thank you for your helpful feedback. The experiments you recommended help clarify the causal relationship between class selectivity and perturbation-robustness.
>
> >Adversarial training...can greatly improve adversarial robustness. If the conclusions in this paper holds, we would expect adversarially trained models to have much better class selectivity than normally trained models. I'm wondering whether the authors could kindly provide these results?
>
> Thank you for this suggestion! Indeed, the causality is bidirectional. We found that PGD training increases the mean level of class selectivity across neurons in a network, and this effect is proportional to the strength of PGD training: networks trained with more strongly-perturbed samples have higher class selectivity (Appendix A.7). This effect was present in both ResNet18 trained on Tiny ImageNet (Figure A16c) and ResNet20 trained on CIFAR10 (Figure A16f), although the effect in ResNet20 is diminished by the presence of dead units (Figure A16d,e}). Controlling for the presence of dead units results in a clear, monotonic effect of PGD training intensity on class selectivity (Figure A16c,f). These results indicate that the causal relationship between worst-case perturbation robustness and class selectivity is bidirectional: increasing class selectivity not only causes increased worst-case perturbation robustness, but increasing worst-case perturbation-robustness also causes increased class selectivity. This result has also been added to the main text (Results 4.2, paragraph 5).
>
> >Similarly, there are some methods (e.g., AugMix [4]) improving model robustness to average-case corruptions. Does AugMix model has significantly lower class selectivity than normally trained models.
>
> The causality is bidirectional here as well! We found that AugMix does indeed decrease the mean level of class selectivity across neurons in a network (Results 4.1, paragraph 4; Appendix A.4; Figure A11). AugMix decreases overall levels of selectivity similarly to training with a class selectivity regularization scale of approximately $\alpha=-0.1$ or $\alpha=-0.2$ in both ResNet18 trained on Tiny ImageNet (Figures A11a-b) and ResNet20 trained on CIFAR10 (Figures A14c-d). These results indicate that the causal relationship between average-case perturbation robustness and class selectivity is bidirectional: not only does decreasing class selectivity cause average-case perturbation robustness to increase, but increasing average-case perturbation-robustness also causes class selectivity to decrease.
>
> >In my point of view, one possible explanation for the different behaviors between average-case and worst-case corruptions could be that normal adversarial images (generated by PGD, FGSM as in your paper) are out-of-distribution samples [3], while average-case corruptions are likely to be on-distribution. So I'm a bit curious about the behaviors of the "on– manifold adversarial examples" defined in [3]: do they behave more like normal adversarial images (e.g., causing a larger increase in early-layer dimensionality) or more like average-case corruptions (causing a smaller increase in early-layer dimensionality )?
>
> We regret that we were unable to perform this analysis in time for the author response deadline, as it is somewhat technically complex (estimating the manifold requires training a GAN-VAE on each data class). We will aim to include it in a final version of the manuscript, should it be accepted.
>
> [1] Morcos et al., On the importance of single directions for generalization. ICLR, 2018.
>
> [2] Madry et al., Towards Deep Learning Models Resistant to Adversarial Attacks. ICLR, 2018.
>
> [3] Stutz et al., Disentangling Adversarial Robustness and Generalization. CVPR, 2019.
>
> [4] Hendrycks et al., AugMix: A Simple Data Processing Method to Improve Robustness and Uncertainty. ICLR, 2020.
>
> [5] Tsipiras et al., Robustness May Be at Odds with Accuracy. ICLR, 2019.
>
> [6] Theoretically Principled Trade-off between Robustness and Accuracy. ICML, 2019.

---

### Official Review · AnonReviewer1 · 2020-10-31
**Many interesting observations regarding robustness, but unclear what conclusions or how to unify these observations**

**Rating:** 6
**Confidence:** 3

**Review:**

This work investigates two classes of perturbation robustness, average-case perturbations which are considered to be naturally occurring in image data, and worst-case perturbations that are perturbations generated by an adversary. Neural network susceptibility to these perturbations is evaluated with respect to a class selectivity metric and dimensionality measure.

The authors find that decreasing class selectivity will increase robustness to average-case perturbations while reducing robustness to worst-case perturbations. Simultaneously, increasing class selectivity improves robustness to worst-case perturbations while reducing average-case perturbation robustness.

In addition, the authors consider the correspondence between the dimensionality of the early layers and observe how this dimensionality corresponds to class selectivity and robustness. They find the dimensionality is inversely related to class selectivity while also positively correlated with reduced worst-case robustness.

While the experiments are thorough in showing a relationship between class selectivity and robustness, many of the observations do not map convincingly to the conclusions. For instance, in Section 4.1, the second paragraph appears to be conjecture which is not justified by the potential observation outcomes. Having higher class selectivity does not necessarily mean fewer potent neurons is the reason for increased average-case robustness. This conclusion is not sufficiently supported by measuring robustness as a function of class selectivity. It also seems odd to hypothesize two different conclusions for opposite outcomes. The same issue of drawing a conclusion regarding the worst-case perturbation analysis in section 4.2. Why create potential conclusions for two opposite outcomes?

The relationship between the definition of an average-case perturbation and a worst-case perturbation is not well-defined. It appears to be a semantic association of what is expected to occur naturally (average-case) and what is caused by an adversary (worst-case). But without some topology or relationship between these two cases, the observations of class selectivity on each are just independent and unrelated observations. For instance, what is in between an average-case and worst-case perturbation? Where do images from datasets like ObjectNet (https://objectnet.dev/) or Natural Adversarial Examples (https://arxiv.org/abs/1907.07174) fall in the average-case to worst-case dimension?

Overall, this work would benefit from a unified view of how class selectivity relations to image perturbations. As it currently stands, this paper appears to be a collection of observations in need of a clear conclusion.

Other notes:
-It would be illustrative to have some visualization of the average-case perturbations.
-$SI_u$ in Eq 3. should also be a function of $l$

---

> ### Author Response · Authors · 2020-11-25
> **Response to reviewer 1**
>
> Thank you for your thoughtful and detailed feedback! Please note that our response is split over two comments due to space limitations.
>
> >While the experiments are thorough in showing a relationship between class selectivity and robustness, many of the observations do not map convincingly to the conclusions. For instance, in Section 4.1, the second paragraph appears to be conjecture which is not justified by the potential observation outcomes. Having higher class selectivity does not necessarily mean fewer potent neurons is the reason for increased average-case robustness. This conclusion is not sufficiently supported by measuring robustness as a function of class selectivity. It also seems odd to hypothesize two different conclusions for opposite outcomes. The same issue of drawing a conclusion regarding the worst-case perturbation analysis in section 4.2. Why create potential conclusions for two opposite outcomes?
>
> We apologize for the lack of clarity. Our goal is to structure our writing in a way that faithfully conveys the hypothesis-driven nature of the scientific process. Because experiments are conducted without knowing the outcomes a priori, we wanted to explicitly articulate at least two possible outcomes for each experiment, and the implications of those outcomes. Doing so avoids a scenario in which all observations are justified post-hoc and conveying the impression that “this is the only way it could have been”. Similarly, after presenting a result, we try to provide the reader with a plausible implication or interpretation of the observed result while avoiding strong claims, for example by using hedges such as “suggests” or “may”.
>
> We had tried to appropriately hedge the interpretation component, but understand your concern that a reader may develop an unjustified level of certainty. Accordingly, we have softened some of the language dealing with interpretation or implication. With regards to “Having higher class selectivity does not necessarily mean fewer potent neurons is the reason for increased average-case robustness” specifically, we felt that this was appropriately contextualized by the discussion of prior research on dimensionality and robustness in the introduction and related work, and fits with our results regarding the relationship between dimensionality, unit stability, and class selectivity.
>
> >The relationship between the definition of an average-case perturbation and a worst-case perturbation is not well-defined. It appears to be a semantic association of what is expected to occur naturally (average-case) and what is caused by an adversary (worst-case). But without some topology or relationship between these two cases, the observations of class selectivity on each are just independent and unrelated observations. For instance, what is in between an average-case and worst-case perturbation? Where do images from datasets like ObjectNet (https://objectnet.dev/) or Natural Adversarial Examples (https://arxiv.org/abs/1907.07174) fall in the average-case to worst-case dimension?
>
> We agree that the relationship between worst-case and average-case perturbations is poorly understood. Indeed, this issue has motivated a number of recent studies attempting to better understand this relationship (e.g. [1-4]), including ours. We also agree that a strong theoretical framework linking worst-case and average-case robustness would be of great value to the field. While this is not part of our work, it is our hope that the results presented here constitute an empirical linkage that serves to advance the field in the same vein as [1-4], which are also primarily empirical.
>
> Where datasets such as ObjectNet and Natural Adversarial Examples fall in the average-case vs. worst-case dimension is an excellent (and open) question. In an attempt to provide empirical evidence addressing this, we tested our models on Natural Adversarial Examples (ImageNet-A; Results 4.2, paragraph 4; Appendix A.6; Figure A15). Interestingly, class selectivity regularization does not appear to affect robustness to natural adversarial examples. Performance on ImageNet-A was similar across all tested class selectivity regularization scales for both ResNet18 (Figure A15a) and ResNet50 (Figure A15b). This indicates that class selectivity regularization may share some limitations with other methods for improving both worst-case and average-case robustness, many of which fail to yield significant robustness improvements against ImageNet-A. Unfortunately we did not have time to conduct experiments on ObjectNet before the author response deadline, but we would certainly include these experiments in the final version of the paper.

---

> > ### Author Response · Authors · 2020-11-25
> > **Response to reviewer 1, cont'd**
> >
> > >Overall, this work would benefit from a unified view of how class selectivity relations to image perturbations. As it currently stands, this paper appears to be a collection of observations in need of a clear conclusion.
> >
> > We will reiterate that we agree that a theoretical framework linking different types of input perturbations would be a boon to interpreting our results, and of immense value to the field more generally. But the core observation of our work can be stated simply: class selectivity imposes a trade-off between average-case vs worst-case perturbation robustness, a trade-off that is mediated by input-unit gradient variability and representational dimensionality. We would also direct you to the statements made by Reviewer 2 about the broader relevance of our work.
> >
> > Furthermore, two new experiments clarify the causal relationship between class selectivity and robustness: models trained with AugMix, which is known to improve average-case robustness [5], causes class selectivity to decrease (Appendix A.4, Figure A11). Furthermore, models trained with PGD-perturbed samples, which confers worst-case robustness, have higher class selectivity (Appendix A.6, Figure A15). This demonstrates that the causal relationship between both kinds of perturbation robustness and class selectivity is bidirectional: decreasing class selectivity increases average-case robustness, and increasing average-case robustness (via AugMix) decreases class selectivity. Likewise, increasing class selectivity increases worst-case robustness, and increasing worst-case robustness (via PGD training) increases class selectivity.
> >
> > Finally, we also feel that our work is practically relevant to the cause of developing robust models: we demonstrate the potential utility of class selectivity as both a metric for measuring model robustness and a method for achieving robustness (via regularization).
> >
> > >It would be illustrative to have some visualization of the average-case perturbations.
> >
> > We provide examples of average-case perturbations in Figure A1, which we now explicitly refer to in multiple locations in the main text. We apologize for the lack of clarity.
> >
> > >SI in Eq 3. should also be a function of l
> >
> > Good find! Fixed, thank you!
> >
> > [1] Ford et al., ICLR, 2019
> >
> > [2] Yin et al., NeurIPS, 2019
> >
> > [3] Dapello et al., NeurIPS, 2020
> >
> > [4] Hendrycks and Dietterich, ICLR, 2019

---

### Author Response · Authors · 2020-11-25
**General response to reviewers**

We appreciate that all four reviewers praised the merits of our work. We are happy to hear that our work addresses an “important topic” (R4). We are also glad that “the experiments are thorough in showing a relationship between class selectivity and robustness” (R1), especially given R2’s enthusiasm about the theoretical and practical implications of our work: “It is good to see the authors generalize [class selectivity’s] usage to measuring model robustness to natural corruptions and adversarial attacks”; “I think this is one step further from the well-know accuracy-robustness tradeoff”. It was also gratifying to hear that the paper was “well-written and easy to follow” (R3), because we wanted readers to come away feeling that that we had shown “interesting empirical findings and exploration of sensitivity on worst-case robustness and average-case robustness” (R3).

We would also like to thank the reviewers for the thoroughness and thoughtfulness of their concerns and suggestions. We conducted a number of new experiments and analyses in an attempt to thoroughly resolve all reviewer concerns that could not be addressed through clarification alone. We summarize the new experiments and analyses here, and describe them in detail in our responses to individual reviewers below.

First, we extended our findings to ResNet50 trained on Tiny ImageNet as per R3’s request to expand the generalizability of our conclusions. We found that effects of selectivity on robustness to average-case and worst-case perturbations are completely consistent with what was observed in our other two models (Figures A9, A10, A13).

We also tested our models on Natural Adversarial Examples (ImageNet-A; Results 4.2, paragraph 4; Appendix A.6; Figure A15), as suggested by R1. Interestingly, class selectivity regularization does not appear to affect robustness to natural adversarial examples in either ResNet18 (Figure A15a) and ResNet50 (Figure A15b). This indicates that class selectivity regularization may share some limitations with other methods for improving both worst-case and average-case robustness, many of which fail to yield significant robustness improvements against ImageNet-A.

We conducted two new experiments to further clarify the causal relationship between class selectivity and robustness to average- and worst-case perturbations. These experiments were suggested by R2, but are also relevant to concerns of R1. First, we found that training with AugMix, which is known to improve average-case robustness, also causes class selectivity to decrease (Results 4.1, paragraph 4; Appendix A.4; Figure A11). Furthermore, we found that PGD training increases the mean level of class selectivity across neurons in a network, and this effect is proportional to the strength of PGD training (Results 4.2, paragraph 5; Appendix A.7; Figure A16). These results indicate that the causal relationship between class selectivity and perturbation robustness is bidirectional: decreasing class selectivity increases average-case robustness, and increasing average-case robustness (via AugMix training) decreases class selectivity. Likewise, increasing class selectivity increases worst-case robustness, and increasing worst-case robustness (via PGD training) increases class selectivity.

We also quantified the intrinsic dimensionality (ID; a non-linear measure of dimensionality) of each layer's representations (Results 4.4, final paragraph; Appendix A.1.4; Figure A22), as suggested by R4. Interestingly, the results were qualitatively similar to what we observed when examining linear dimensionality (Figure A22), and was present in both ResNet18 trained on Tiny ImageNet (Figure A22a-c) and ResNet20 trained on CIFAR10 (Figure A22d-f). Thus both linear and non-linear measures of dimensionality imply that representational dimensionality may present a trade-off between worst-case and average-case perturbation robustness.

Thank you,

Authors

---

### Decision · Program_Chairs · 2021-01-07
**Final Decision**

**Decision:**

Reject

**Comment:**

The reviewers had raised a number of concerns which were mostly addressed during the discussion phase thanks to the additional experiments/explanations that the authors provided. However, some of the reviewers are not yet convinced about the main claims of the paper. While the paper provides a number of interesting/important/novel observations, which are obtained using an extensive set of carefully designed experiments, there are still some ambiguities about the main claims that may need further investigation/ justification.

One general issue with the current version is lack of clarity. I recommend that the authors revise the writing of the paper and make an effort to better explain/justify the main claims, ideas, and setups in the paper. Perhaps it would help to add a section early in the paper and define/review the basic concepts/definitions.  I also encourage the authors to reason about their main claims, choice of loss function (and why it is the right choice), and their experimental setups.   As an example, the authors should further investigate the impact of their proposed regularization on semantic representations.  The authors interpret the (distribution of) class information as “semantic” content/features. Indeed, when considering data with a variety of average-case perturbations,  one could argue that semantic features like brightness or snow are actually uniform over classes, which might means that class selectivity could not appropriately capture the effect of those features. To evaluate semantic robustness,   it seems necessary to find a way to isolate specific semantic features in the data, i.e. by changing from snow to rain in corresponding images, rather than looking at the performance on the same brightness subset for different levels of their regularization parameter alpha.  Hence there is a need for further investigation on the effectiveness of class selectivity.

Some of the reviewers have indicated that differences in the class selectivity curves in Figures 1,2 appear marginal (e.g. at most 3% difference). Hence, additional experiments (with other data sets) could be beneficial in this regard.